# Efficient Model Updates for Approximate Unlearning of Graph-Structured Data

**Eli Chien**[*]    **Chao Pan**[*]    **Olgica Milenkovic**
Department of Electrical and Computer Engineering
University of Illinois, Urbana-Champaign
{ichien3,chaopan2,milenkov}@illinois.edu

## Abstract

With the adoption of recent laws ensuring the "right to be forgotten", the problem of machine unlearning has become of significant importance. This is particularly the case for graph-structured data, and learning tools specialized for such data, including graph neural networks (GNNs). This work introduces the first known approach for *approximate graph unlearning* with provable theoretical guarantees. The challenges in addressing the problem are two-fold. First, there exist multiple different types of unlearning requests that need to be considered, including node feature, edge and node unlearning. Second, to establish provable performance guarantees, one needs to carefully evaluate the process of feature mixing during propagation. We focus on analyzing Simple Graph Convolutions (SGC) and their generalized PageRank (GPR) extensions, thereby laying the theoretical foundations for unlearning GNNs. Empirical evaluations of six benchmark datasets demonstrate excellent performance/complexity/privacy trade-offs of our approach compared to complete retraining and general methods that do not leverage graph information. For example, unlearning $200$ out of $1208$ training nodes of the Cora dataset only leads to a $0.1\%$ loss in test accuracy, but offers a $4$-fold speed-up compared to complete retraining with a $(\epsilon, \delta) = (1, 10^{-4})$ "privacy cost". We also exhibit a $12\%$ increase in test accuracy for the same dataset when compared to unlearning methods that do not leverage graph information, with comparable time complexity and the same privacy guarantee. Our code is available online[1].

## 1 Introduction

Machine learning algorithms are used in many application domains, including biology, computer vision and natural language processing. Relevant models are often trained either on third-party datasets, internal or customized subsets of publicly available user data. For example, many computer vision models are trained on images from Flickr users (Thomee et al., 2016; Guo et al., 2020) while many natural language processing (e.g., sentiment analysis) and recommender systems heavily rely on repositories such as IMDB (Maas et al., 2011). Furthermore, numerous ML classifiers in computational biology are trained on data from the UK Biobank (Sudlow et al., 2015), which represents a collection of genetic and medical records of roughly half a million participants (Ginart et al., 2019). With recent demands for increased data privacy, the above referenced and many other data repositories are facing increasing demands for data removal. Certain laws are already in place guaranteeing the rights of certified data removal, including the European Union's General Data Protection Regulation (GDPR), the California Consumer Privacy Act (CCPA) and the Canadian Consumer Privacy Protection Act (CPPA) (Sekhari et al., 2021).

Removing user data from a dataset is insufficient to guarantee the desired level of privacy, since models trained on the original data may still contain information about their patterns and features. This consideration gave rise to a new research direction in machine learning, referred to as *machine unlearning* (Cao & Yang, 2015), in which the goal is to guarantee that the user data information is also removed from the trained model. Naively, one can retrain the model from scratch to meet

---

[*]Equal contribution.
[1]https://github.com/thupchnsky/sgc_unlearn

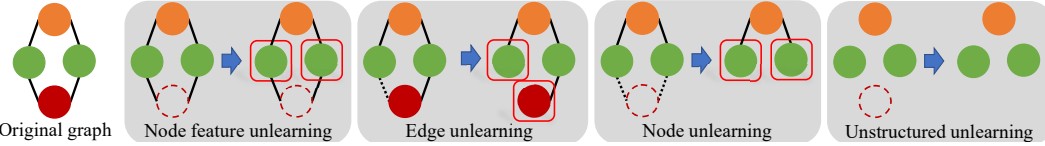

Figure 1: Illustration of three different types of approximate graph unlearning problems and a comparison with the case of unlearning without graph information (Guo et al., 2020). The colors of the nodes capture properties of node features, and the red frame indicates node embeddings affected by 1-hop propagation. When no graph information is used, the node embeddings are uncorrelated. However, for the case of graph unlearning problems, removing one node or edge can affect the node embeddings of the entire graph for a large enough number of propagation steps.

the privacy demand, yet retraining comes at a high computation cost and is thus not practical when accommodating frequent removal requests. To avoid complete retraining, various methods for machine unlearning have been proposed, including *exact approaches* (Ginart et al., 2019; Bourtoule et al., 2021) as well as *approximate methods* (Guo et al., 2020; Sekhari et al., 2021).

At the same time, graph-centered machine learning has received significant interest from the learning community due to the ubiquity of graph-structured data. Usually, the data contains two sources of information: Node features and graph topology. Graph Neural Networks (GNN) leverage both types of information simultaneously and achieve state-of-the-art performance in numerous real-world applications, including Google Maps (Derrow-Pinion et al., 2021), various recommender system (Ying et al., 2018), self-driving cars (Gao et al., 2020) and bioinformatics (Zhang et al., 2021b). Clearly, user data is involved in training the underlying GNNs and it may therefore be subject to removal. However, it is still unclear how to perform unlearning of GNNs.

We take the first step towards solving the approximate unlearning problem by performing a nontrivial theoretical analysis of some simplified GNN architectures. Inspired by the unstructured data *certified removal procedure* (Guo et al., 2020), we propose the first known approach for *approximate graph unlearning*. Our main contributions are as follows. First, we introduce three types of data removal requests for graph unlearning: Node feature unlearning, edge unlearning and node unlearning (see Figure 1). Second, we derive theoretical guarantees for approximate graph unlearning mechanisms for all three removal cases on SGC (Wu et al., 2019) and their GPR generalizations. In particular, we analyze $L_2$-regularized graph models trained with differentiable convex loss functions. The analysis is challenging since propagation on graphs "mixes" node features. Our analysis reveals that the degree of the unlearned node plays an important role in the unlearning process, while the number of propagation steps may or may not be important for different unlearning scenarios. To the best of our knowledge, the theoretical guarantees established in this work are the first provable approximate unlearning studies for graphs. Furthermore, the proposed analysis also encompasses node classification and node regression problems. Third, our empirical investigation on frequently used datasets for GNN learning shows that our method offers an excellent performance-complexity-privacy trade-off. For example, when unlearning 200 out of 1208 training nodes of the Cora dataset, our method offers comparable test accuracy as complete retraining, but offers a 4-fold speed-up with a $(\epsilon, \delta) = (1, 10^{-4})$ "privacy cost". We also test our model on datasets for which removal requests are most likely to arise, including Amazon co-purchase networks.

Due to space limitations, all proofs and some detailed discussions are relegated to the Appendix.

## 2 RELATED WORKS

**Machine unlearning and certified data removal.** Cao & Yang (2015) introduced the concept of machine unlearning and proposed distributed learners for exact unlearning. Bourtoule et al. (2021) introduced sharding-based methods for unlearning, while Ginart et al. (2019) described unlearning approaches for $k$-means clustering. These works focused on *exact unlearning*: The unlearned model is required to perform identically to a completely retrained model. As an alternative, Guo et al. (2020) introduced a probabilistic definition of unlearning motivated by differential privacy (Dwork, 2011). Sekhari et al. (2021) studied the generalization performance of machine unlearning methods.

Golatkar et al. (2020) proposed heuristic-based selective forgetting in deep networks. These probabilistic approaches naturally allow for "approximate" unlearning. None of these works addressed the machine unlearning problem on graphs. To the best of our knowledge, the only work in this direction is GraphEraser (Chen et al., 2021). However, the strategy proposed therein uses sharding, which only works for exact unlearning and is hence completely different from our approximate approach. Also, the approach in Chen et al. (2021) relies on partitioning the graph using community detection methods. It therefore implicitly makes the assumption that the graph is homophilic which is not warranted in practice (Chien et al., 2021b; Lim et al., 2021). In contrast, our method works for arbitrary graphs and allows for approximate unlearning while ensuring excellent trade-offs among performance, privacy and complexity.

**Differential privacy (DP) and DP-GNNs.** Machine unlearning, especially the approximation version described in Guo et al. (2020), is closely related to differential privacy (Dwork, 2011). In fact, differential privacy is a *sufficient condition* for machine unlearning. If a model is differentially private, then the adversary cannot distinguish whether the model is trained on the original dataset or on a dataset in which one data point is removed. Hence, even *without* model updating, a DP model will automatically unlearn the removed data point (see also the explanation in (Ginart et al., 2019; Sekhari et al., 2021) and Figure 4). Although DP is a sufficient condition for unlearning, it is not a necessary condition. Also, most of the DP models suffer from a significant degradation in performance even when the privacy constraint is loose (Chaudhuri et al., 2011; Abadi et al., 2016). Machine unlearning can therefore be viewed as a means to trade-off between performance and computational cost, with complete retraining and DP on two different ends of the spectrum (Guo et al., 2020). Several recent works proposed DP-GNNs (Daigavane et al., 2021; Olatunji et al., 2021a; Wu et al., 2021; Sajadmanesh et al., 2022) – however, even for unlearning *one single node or edge*, these methods require a high "privacy cost" ($\epsilon \geq 5$) to learn with sufficient accuracy.

**Graph neural networks.** While GNNs are successfully used for many graph-related problems, accompanying theoretical analyses are usually difficult due to the combination of nonlinear feature transformation and graph propagation. Recently, several simplified GNN models were proposed that can further the theoretical understanding of their performance and scalability. SGCs (Wu et al., 2019) simplify Graph Convolutional Networks (GCNs) (Kipf & Welling, 2017) via linearization (i.e., through the removal of all nonlinearities); although SGC in general underperforms compared to state-of-the-art GNNs, they still offer competitive performance on many datasets. The analysis of SGCs elucidated the relationship between low-pass graph filtering and GCNs which reveals both advantages and potential limitations of GNNs. The GPR generalization of SGC is closely related to many important models that resolve different issues inherent to GNNs. For example, GPRGNN (Chien et al., 2021b) addresses the problem of universal learning on homophilic and heterophilic graph datasets and the issue of over-smoothing. SIGN (Frasca et al., 2020) based graph models and $S^2GC$ (Zhu & Koniusz, 2020) allow for arbitrary sized mini-batch training, which improves the scalability and leads to further performance improvements of methods (Sun et al., 2021; Zhang et al., 2021a; Chien et al., 2022a) on the Open Graph Benchmark leaderboard Hu et al. (2020). Hence, developing approximate graph unlearning approaches for SGC and generalizations thereof is not only of theoretical interest, but also of practical importance.

## 3 PRELIMINARIES

**Notation.** We reserve bold-font capital letters such as $\mathbf{S}$ for matrices and bold-font lowercase letters such as $\mathbf{s}$ for vectors. We use $\mathbf{e}_i$ to denote the $i^{th}$ standard basis, so that $\mathbf{e}_i^T \mathbf{S}$ and $\mathbf{S} \mathbf{e}_i$ represent the $i^{th}$ row and column vector of $\mathbf{S}$, respectively. The absolute value $|\cdot|$ is applied component-wise on both matrices and vectors. We also use the symbols $\mathbf{1}$ for the all-one vector and $\mathbf{I}$ for the identity matrix. Furthermore, we let $\mathcal{G} = (\mathcal{V}, \mathcal{E})$ stand for an undirected graph with node set $\mathcal{V} = [n]$ of size $n$ and edge set $\mathcal{E}$. The symbols $\mathbf{A}$ and $\mathbf{D}$ are used to denote the corresponding adjacency and node degree matrix, respectively. The feature matrix is denoted by $\mathbf{X} \in \mathbb{R}^{n \times F}$ and the features have dimension $F$; For binary classification, the label are summarized in $\mathbf{Y} \in \{-1, 1\}^n$, while the nonbinary case is discussed in Section 5. The relevant norms are $\|\cdot\|$, the $l_2$ norm, and $\|\cdot\|_F$, the Frobenius norm. Note that we use $\|\cdot\|$ for both row and column vectors to simplify the notation. The matrices $\mathbf{A}$ and $\mathbf{D}$ should not be confused with the symbols for an algorithm $A$ and dataset $\mathcal{D}$.

**Certified removal.** Let $A$ be a (randomized) learning algorithm that trains on $\mathcal{D}$, the set of data points before removal, and outputs a model $h \in \mathcal{H}$, where $\mathcal{H}$ represents a chosen space of models. The removal of a subset of points from $\mathcal{D}$ results in $\mathcal{D}'$. For instance, let $\mathcal{D} = (\mathbf{X}, \mathbf{Y})$. Suppose we want to remove a data point, $(\mathbf{e}_i^T \mathbf{X}, \mathbf{e}_i^T \mathbf{Y})$ from $\mathcal{D}$, resulting in $\mathcal{D}' = (\mathbf{X}', \mathbf{Y}')$. Here, $\mathbf{X}', \mathbf{Y}'$ are equal to $\mathbf{X}, \mathbf{Y}$, respectively, except that the row corresponding to the removed data point is deleted. Given $\epsilon > 0$, an unlearning algorithm $M$ applied to $A(\mathcal{D})$ is said to guarantee an $(\epsilon, \delta)$-certified removal for $A$, where $\epsilon, \delta > 0$ and $\mathcal{X}$ denotes the space of possible datasets, if

$$\forall \mathcal{T} \subseteq \mathcal{H}, \mathcal{D} \subseteq \mathcal{X}, i \in [n]: \ \mathbb{P}\left(M(A(\mathcal{D}), \mathcal{D}, \mathcal{D} \setminus \mathcal{D}') \in \mathcal{T}\right) \leq \exp(\epsilon) \mathbb{P}\left(A(\mathcal{D}') \in \mathcal{T}\right) + \delta,$$
$$\mathbb{P}\left(A(\mathcal{D}') \in \mathcal{T}\right) \leq \exp(\epsilon) \mathbb{P}\left(M(A(\mathcal{D}), \mathcal{D}, \mathcal{D} \setminus \mathcal{D}') \in \mathcal{T}\right) + \delta. \tag{1}$$

This definition is related to $(\epsilon, \delta)$-DP (Dwork, 2011) except that we are allowed to update the model based on the removed point (see Figure 4). An $(\epsilon, \delta)$-certified removal method guarantees that the updated model $M(A(\mathcal{D}), \mathcal{D}, \mathcal{D} \setminus \mathcal{D}')$ is "approximately" the same as the model $A(\mathcal{D}')$ obtained by retraining from scratch. Thus, any information about the removed data $\mathcal{D} \setminus \mathcal{D}'$ is "approximately" eliminated from the model. Ideally, we would like to design $M$ such that it satisfies equation (1) and has a complexity that is significantly smaller than that of complete retraining.

## 4 Approximate Graph Unlearning with Theoretical Guarantees

Unlike standard machine unlearning, approximate graph unlearning uses datasets that contain not only node features $\mathbf{X}$ but also the graph topology $\mathbf{A}$, and therefore require different data removal procedures. We focus on node classification, for which the training dataset equals $\mathcal{D} = (\mathbf{X}, \mathbf{Y}_{T_r}, \mathbf{A})$. Here, $\mathbf{Y}_{T_r}$ is identical to $\mathbf{Y}$ on rows indexed by points of the training set $T_r$ while the remaining rows are all zeros. Without loss of generality, we assume that the training set comprises the first $m$ nodes (i.e. $T_r = [m]$), where $m \leq n$. An unlearning method $M$ achieves $(\epsilon, \delta)$-approximate graph unlearning with algorithm $A$ if equation 1 is satisfied for $\mathcal{D} = (\mathbf{X}, \mathbf{Y}_{T_r}, \mathbf{A})$ and $\mathcal{D}'$, which differ based on the type of requests: Node feature unlearning, edge unlearning, and node unlearning.

### 4.1 Unlearning SGC and comparison with unstructured unlearning

SGC is a simplification of GCN obtained by removing all nonlinearities from the latter model. This leads to the following update rule: $\mathbf{P}^K \mathbf{X} \mathbf{W} \triangleq \mathbf{Z} \mathbf{W}$, where $\mathbf{W}$ denotes the matrix of learnable weights, $K \geq 0$ equals the number of propagation steps and $\mathbf{P}$ denotes the one-step propagation matrix. The standard choice of the propagation matrix is the symmetric normalized adjacency matrix with self-loops, $\mathbf{P} = \tilde{\mathbf{D}}^{-1/2} \tilde{\mathbf{A}} \tilde{\mathbf{D}}^{-1/2}$, where $\tilde{\mathbf{A}} = \mathbf{A} + \mathbf{I}$ and $\tilde{\mathbf{D}}$ equals the degree matrix with respect to $\tilde{\mathbf{A}}$. We will work with the asymmetric normalized version of $\mathbf{P}$, $\mathbf{P} = \tilde{\mathbf{D}}^{-1} \tilde{\mathbf{A}}$. This choice is made purely for analytical purposes and our empirical results confirm that this normalization ensures the competitive performance of our unlearning methods.

The resulting node embedding is used for node classification by choosing an appropriate loss (i.e., logistic loss) and minimizing the $L_2$-regularized empirical risk. For binary classification, $\mathbf{W}$ can be replaced by a vector $\mathbf{w}$; the loss equals $L(\mathbf{w}, \mathcal{D}) = \sum_{i: \mathbf{e}_i^T \mathbf{Y}_{T_r} \neq 0} \left( \ell(\mathbf{e}_i^T \mathbf{Z} \mathbf{w}, \mathbf{e}_i^T \mathbf{Y}_{T_r}) + \frac{\lambda}{2} \|\mathbf{w}\|^2 \right)$, where $\ell(\mathbf{e}_i^T \mathbf{Z} \mathbf{w}, \mathbf{e}_i^T \mathbf{Y}_{T_r})$ is a convex loss function that is differentiable everywhere. We also write $\mathbf{w}^\star = A(\mathcal{D}) = \arg\min_{\mathbf{w}} L(\mathbf{w}, \mathcal{D})$, where the optimizer is unique whenever $\lambda > 0$.

We start with a high-level description of the approximate unstructured unlearning approach introduced in Guo et al. (2020). Note that "certified removal" in the context of the former work refers to *approximate unlearning* that provably satisfies (1). Let us denote the Hessian of $L(\cdot; \mathcal{D}')$ at $\mathbf{w}^\star$ by $\mathbf{H}_{\mathbf{w}^\star} = \nabla^2 L(\mathbf{w}^\star; \mathcal{D}')$. The authors of Guo et al. (2020) propose the following mechanism for unlearning the $m^{th}$ training point: $\mathbf{w}^- = M(\mathbf{w}^\star, \mathcal{D}, \mathcal{D} \setminus \mathcal{D}') = \mathbf{w}^\star + \mathbf{H}_{\mathbf{w}^\star}^{-1} \Delta_{guo}$, where $\Delta_{guo} = \lambda \mathbf{w}^\star + \nabla \ell(\mathbf{e}_m^T \mathbf{X} \mathbf{w}^\star, \mathbf{e}_m^T \mathbf{Y}_{T_r})$. When $\nabla L(\mathbf{w}^-, \mathcal{D}') = 0$, then $\mathbf{w}^-$ is the unique optimizer of $L(\cdot; \mathcal{D}')$. If $\nabla L(\mathbf{w}^-, \mathcal{D}') \neq 0$, then information about the removed data point remains present in the model. One can show that the gradient residual norm $\|\nabla L(\mathbf{w}^-, \mathcal{D}')\|$ determines the error of $\mathbf{w}^-$ when used to approximate the true minimizer of $L(\cdot; \mathcal{D}')$ (Guo et al., 2020). Hence, upper bounds on $\|\nabla L(\mathbf{w}^-, \mathcal{D}')\|$ can be used to establish approximate unlearning guarantees. More precisely, assume that we have $\|\nabla L(\mathbf{w}^-, \mathcal{D}')\| \leq \epsilon'$ for some $\epsilon' > 0$. Furthermore, consider training with the noisy loss $L_\mathbf{b}(\mathbf{w}, \mathcal{D}) = \sum_{i: \mathbf{e}_i^T \mathbf{Y}_{T_r} \neq 0} \left( \ell(\mathbf{e}_i^T \mathbf{X} \mathbf{w}, \mathbf{e}_i^T \mathbf{Y}_{T_r}) + \frac{\lambda}{2} \|\mathbf{w}\|^2 \right) + \mathbf{b}^T \mathbf{w}$, where $\mathbf{b}$ is drawn randomly according to some distribution. Then one can leverage the following result.

**Theorem 4.1** (Theorem 3 from Guo et al. (2020)). *Let A be the learning algorithm that returns the unique optimum of the loss $L_{\mathbf{b}}(\mathbf{w}, \mathcal{D})$. Suppose that $\|\nabla L_{\mathbf{b}}(\mathbf{w}^-, \mathcal{D}')\| \le \epsilon'$, for some computable bound $\epsilon' > 0$ independent on $\mathbf{b}$ and achieved by $M$. If $\mathbf{b} \sim \mathcal{N}(0, (c_0 \epsilon'/\epsilon)^2)^d$ with $c_0 > 0$, then $M$ satisfies (1) with parameters $(\epsilon, \delta)$ for algorithm A applied to $\mathcal{D}'$, where $\delta = 1.5e^{-c_0^2/2}$.*

Hence, if we can prove that $\|\nabla L(\mathbf{w}^-, \mathcal{D}')\|$ is appropriately bounded for the graph setting as well, then the unlearning mechanism $M$ will ensure $(\epsilon, \delta)$-approximate graph unlearning. One of the main contributions of Guo et al. (2020) is to bound the gradient residual norm $\|\nabla L(\mathbf{w}^-, \mathcal{D}')\|$ of the proposed unlearning mechanism with $\Delta_{guo} = \lambda \mathbf{w}^\star + \nabla \ell(\mathbf{e}_m^T \mathbf{X} \mathbf{w}^\star, \mathbf{e}_m^T \mathbf{Y}_{T_r})$.

Motivated by the unlearning approach from Guo et al. (2020) pertaining to unstructured data, we design an unlearning mechanism for graphs. We generalize their unlearning mechanism by replacing $\Delta_{guo}$ with $\Delta = \nabla L(\mathbf{w}^\star, \mathcal{D}) - \nabla L(\mathbf{w}^\star, \mathcal{D}')$. As an demonstrative example, for node unlearning we consequently have

$$\Delta = \lambda \mathbf{w}^\star + \nabla \ell(\mathbf{e}_m^T \mathbf{Z} \mathbf{w}^\star, \mathbf{e}_m^T \mathbf{Y}_{T_r}) + \sum_{i=1}^{m-1} \left[ \nabla \ell(\mathbf{e}_i^T \mathbf{Z} \mathbf{w}^\star, \mathbf{e}_i^T \mathbf{Y}_{T_r}) - \nabla \ell(\mathbf{e}_i^T \mathbf{Z}' \mathbf{w}^\star, \mathbf{e}_i^T \mathbf{Y}_{T_r}) \right]. \quad (2)$$

Note that our generalized unlearning mechanism matches that ot Guo et al. (2020) when no graph information is present. This can be seen by setting $K = 0$, which leads to $\mathbf{Z} = \mathbf{X}$ and $\mathbf{e}_i^T \mathbf{Z} = \mathbf{e}_i^T \mathbf{Z}' \; \forall i \in [m-1]$. Hence, the third term in equation (2) is zero and thus $\Delta = \Delta_{guo}$. Note that when graph information is present, the third term in equation (2) is, in general, bounded away from zero. This term captures the impact of unlearning node $m$ on all remaining training nodes $[m-1]$, and including effects pertaining to *edge* and *feature* removal. This not only highlights the necessity of investigating generalized unlearning mechanisms, but also the main difficulty of extending the analysis of Guo et al. (2020) to graphs. A more detailed discussion regarding the intuition behind our approach can be found in Appendix A.3. The main technical contribution of our work is to establish bounds of the gradient residual norm for all three types of graph unlearning scenarios. For this analysis, we need the loss function $\ell$ to satisfy the following properties.

**Assumption 4.2.** *For any $\mathcal{D}$, $i \in [n]$ and $\mathbf{w} \in \mathbb{R}^F$: (1) $\|\nabla \ell(\mathbf{e}_i^T \mathbf{Z} \mathbf{w}, \mathbf{e}_i^T \mathbf{Y})\| \le c$ (i.e. the norm of $\nabla \ell$ is c-bounded); (2) $\ell''$ is $\gamma_2$-Lipschitz; (3) $\|\mathbf{e}_i^T \mathbf{X}\| \le 1$; (4) $\ell'$ is $\gamma_1$-Lipschitz; (5) $\ell'$ is $c_1$-bounded.*

Assumptions (1)-(3) are also needed for unstructured unlearning of linear classifiers (Guo et al., 2020). To account for graph-structured data, we require additional assumptions (4)-(5) to establish worst-case bounds. The additional assumptions may be avoided when working with data-dependent bounds (Section 5). In all subsequent derivations, we assume that the unlearned data point corresponds to the $m^{th}$ node for node feature and node unlearning; for edge unlearning, we wish to unlearn the edge $(1, m)$. Generalizations for multiple unlearning requests are discussed in Section 5.

## 4.2 Node feature unlearning for SGCs

We start with the simplest type of unlearning – node feature unlearning – for SGCs. In this case, we remove the node feature and label of one node from $\mathcal{D}$, resulting in $\mathcal{D}' = (\mathbf{X}', \mathbf{Y}'_{T_r}, \mathbf{A})$. The matrices $\mathbf{X}', \mathbf{Y}'_{T_r}$ are identical to $\mathbf{X}, \mathbf{Y}_{T_r}$, respectively, except for the $m^{th}$ row of the former being set to zero. Note that in this case, the graph structure remains unchanged.

**Theorem 4.3.** *Suppose that Assumption 4.2 holds. For the node feature unlearning scenario and $\mathbf{Z} = \mathbf{P}^K \mathbf{X}$ and $\mathbf{P} = \tilde{\mathbf{D}}^{-1} \tilde{\mathbf{A}}$, we have*

$$\|\nabla L(\mathbf{w}^-, \mathcal{D}')\| = \|(\mathbf{H}_{\mathbf{w}_\eta} - \mathbf{H}_{\mathbf{w}^\star}) \mathbf{H}_{\mathbf{w}^\star}^{-1} \Delta\| \le \frac{\gamma_2 (2c\lambda + (c\gamma_1 + \lambda c_1) \tilde{\mathbf{D}}_{mm})^2}{\lambda^4 (m-1)}, \quad (3)$$

where $\mathbf{H}_{\mathbf{w}_\eta}$ denotes the Hessian of $L(\cdot; \mathcal{D}')$ at $\mathbf{w}_\eta = \mathbf{w}^\star + \eta \mathbf{H}_{\mathbf{w}^\star}^{-1} \Delta$ for some $\eta \in [0, 1]$. A similar conclusion holds for the case when we wish to unlearn node features of a node that is not in $T_r$. In this case we just replace $\tilde{\mathbf{D}}_{mm}$ by the degree of the corresponding node. This result shows that the norm bound is large if the unlearned node has a large degree, since a large-degree node will affect the values of many rows in $\mathbf{Z}$. Our result also demonstrates that the norm bound is *independent* of $K$, due to the fact that $\mathbf{P}$ is right stochastic. We provide next a sketch of the proof to illustrate the analytical challenges of graph unlearning compared to those of unstructured data unlearning.

Although for node feature unlearning the graph topology does not change, all rows of $\mathbf{Z} = \mathbf{P}^K \mathbf{X}$ may potentially change due to graph information propagation. Thus, the original analysis from (Guo et al., 2020), which corresponds to the special case $\mathbf{Z} = \mathbf{X}$, cannot be applied directly. There are two particular challenges. The first is to ensure that the norm of each row of $\mathbf{Z}$ is bounded by 1. We provide Lemma A.1 to guarantee this. It is critical to choose $\mathbf{P} = \tilde{\mathbf{D}}^{-1} \tilde{\mathbf{A}}$ since all other choices of degree normalization lead to worse bounds (see Appendix A.11). The second and more difficult challenge is to bound $\|\Delta\|$. When $\mathbf{Z} = \mathbf{X}$, the third term in equation (2) is exactly zero, in accordance with Guo et al. (2020). Due to graph propagation, we have to further bound the norm of the third term, which is highly nontrivial since the upper bound is not allowed to grow with $m$ or $n$. We first focus on one of the $m - 1$ terms in the sum. Using Assumption 4.2, one can bound this term by $\|\mathbf{e}_i^T(\mathbf{Z} - \mathbf{Z}')\|$ (we suppressed the dependency on $\lambda, c, c_1$ and $\gamma_1$ for simplicity). The key analytical novelty is to explore the sparsity of $\mathbf{Z} - \mathbf{Z}' = \mathbf{P}^K(\mathbf{X} - \mathbf{X}')$. Note that $\mathbf{X} - \mathbf{X}'$ is an all-zero matrix except for its $m^{th}$ row being equal to $\mathbf{e}_m^T \mathbf{X}$. Thus, we have $\|\mathbf{e}_i^T(\mathbf{Z} - \mathbf{Z}')\| = \|\mathbf{e}_i^T \mathbf{P}^K(\mathbf{X} - \mathbf{X}')\| = \|\mathbf{e}_i^T \mathbf{P}^K \mathbf{e}_m \mathbf{e}_m^T \mathbf{X}\| \le \mathbf{e}_i^T \mathbf{P}^K \mathbf{e}_m$, where the last bound follows from the Cauchy-Schwartz inequality, (3) in Assumption 4.2 and the fact that $\mathbf{P}^K$ is a (component-wise) nonnegative matrix. Thus, summing over $i \in [m - 1]$ leads to the upper bound $\mathbf{1}^T \mathbf{P}^K \mathbf{e}_m$, since $m \le n$. Next, observe that $\mathbf{1}^T \mathbf{P}^K \mathbf{e}_m = \mathbf{1}^T \mathbf{P}^K \tilde{\mathbf{D}}^{-1} \tilde{\mathbf{D}} \mathbf{e}_m = \mathbf{1}^T \left( \tilde{\mathbf{D}}^{-1} \tilde{\mathbf{A}} \right)^K \tilde{\mathbf{D}}^{-1} \mathbf{e}_m \tilde{\mathbf{D}}_{mm} = \mathbf{1}^T \tilde{\mathbf{D}}^{-1} \left( \tilde{\mathbf{A}} \tilde{\mathbf{D}}^{-1} \right)^K \mathbf{e}_m \tilde{\mathbf{D}}_{mm}$. Since $\tilde{\mathbf{A}} \tilde{\mathbf{D}}^{-1}$ is a left stochastic matrix, $\tilde{\mathbf{A}} \tilde{\mathbf{D}}^{-1} \mathbf{p}$ is a probability vector whenever $\mathbf{p}$ is a probability vector. Clearly, $\mathbf{e}_m$ is a probability vector. Hence, $(\tilde{\mathbf{A}} \tilde{\mathbf{D}}^{-1})^K \mathbf{e}_m$ is also a probability vector. Since all diagonal entries of $\tilde{\mathbf{D}}^{-1}$ are nonnegative and upper bounded by 1 given the self-loops for all nodes, $\mathbf{1}^T \tilde{\mathbf{D}}^{-1} \mathbf{p} \le \mathbf{1}^T \mathbf{p} = 1$ for any probability vector $\mathbf{p}$. Hence, the term above is bounded by $\tilde{\mathbf{D}}_{mm}$. The bound depends on $\tilde{\mathbf{D}}_{mm}$ and does not increase with $m$ or $K$. Although node feature unlearning is the simplest case of graph unlearning, our sketch of the proof illustrates the difficulties associated with bounding the third term in $\Delta$. Similar, but more complicated approaches are needed for the analysis of edge unlearning and node unlearning.

## 4.3 Edge and node unlearning for SGCs and GPRs Extensions

**Edge unlearning for SGC.** We describe next the bounds for edge unlearning and highlight the technical issues arising in the analysis of this setting. Here, we remove one edge $(1, m)$ from $\mathcal{D}$, resulting in $\mathcal{D}' = (\mathbf{X}, \mathbf{Y}_{T_r}, \mathbf{A}')$. The matrix $\mathbf{A}'$ is identical to $\mathbf{A}$ except for $\tilde{\mathbf{A}}'_{1m} = \tilde{\mathbf{A}}'_{m1} = 0$. Furthermore, $\tilde{\mathbf{D}}'$ is the degree matrix corresponding to $\tilde{\mathbf{A}}'$. Note that the node features and labels remain unchanged.

**Theorem 4.4.** *Suppose that Assumption 4.2 holds. Under the edge unlearning scenario, and for* $\mathbf{P} = \tilde{\mathbf{D}}^{-1} \tilde{\mathbf{A}}$ *and* $\mathbf{Z} = \mathbf{P}^K \mathbf{X}$, *we have*

$$\|\nabla L(\mathbf{w}^-, \mathcal{D}')\| = \|(\mathbf{H}_{\mathbf{w}_\eta} - \mathbf{H}_{\mathbf{w}^\star})\mathbf{H}_{\mathbf{w}^\star}^{-1} \Delta\| \le \frac{16\gamma_2 K^2 \left(c\gamma_1 + c_1\lambda\right)^2}{\lambda^4 m}. \tag{4}$$

Similar to what holds for the node feature unlearning case, Theorem 4.4 still holds when neither of the two end nodes of the removed edge belongs to $T_r$.

**Node unlearning for SGC.** We now discuss the most difficult case, node unlearning. In this case, one node is entirely removed from $\mathcal{D}$, including node features, labels and edges. This results in $\mathcal{D}' = (\mathbf{X}', \mathbf{Y}'_{T_r}, \mathbf{A}')$. The matrices $\mathbf{X}', \mathbf{Y}'_{T_r}$ are defined similarly to those described for node feature unlearning. The matrix $\mathbf{A}'$ is obtained by replacing the $m^{th}$ row and column in $\mathbf{A}$ by all-zeros (similar changes are introduced in $\tilde{\mathbf{A}}$, with $\tilde{\mathbf{A}}_{mm} = 0$). For simplicity, we let $\tilde{\mathbf{D}}'_{mm} = 1$ as this assumption does not affect the propagation results.

**Theorem 4.5.** *Suppose that Assumption 4.2 holds. For the node unlearning scenario and* $\mathbf{Z} = \mathbf{P}^K \mathbf{X}$ *and* $\mathbf{P} = \tilde{\mathbf{D}}^{-1} \tilde{\mathbf{A}}$, *we have*

$$\|\nabla L(\mathbf{w}^-, \mathcal{D}')\| = \|(\mathbf{H}_{\mathbf{w}_\eta} - \mathbf{H}_{\mathbf{w}^\star})\mathbf{H}_{\mathbf{w}^\star}^{-1} \Delta\| \le \frac{\gamma_2 \left(2c\lambda + K\left(c\gamma_1 + c_1\lambda\right)\left(2\tilde{\mathbf{D}}_{mm} - 1\right)\right)^2}{\lambda^4(m-1)}. \tag{5}$$

The main challenge arises in the proof of Theorem 4.4 and 4.5 is bounding $\|\Delta\|$ appropriately. Unlike for the node feature unlearning case, now both graph structure and node features can change

due to the unlearning request. We establish a series of lemmas to characterize the difference between $\mathbf{Z}$ and $\mathbf{Z}'$, which play important roles in our proofs (see Appendix A.8 and A.9 for complete proofs).

**Approximate graph unlearning in GPR-based model.** Our analysis can be extended to Generalized PageRank (GPR)-based models (Li et al., 2019). The definition of GPR is $\sum_{k=0}^{K} \theta_k \mathbf{P}^k \mathbf{S}$, where $\mathbf{S}$ denotes a node feature or node embedding. The learnable weights $\theta_k$ are called GPR weights and different choices for the weights lead to different propagation rules (Jeh & Widom, 2003; Chung, 2007). GPR-type propagations include SGC and APPNP rules as special cases (Chien et al., 2021b). If we use linearly transformed features $\mathbf{S} = \mathbf{X}\bar{\mathbf{W}}$, for some weight matrix $\bar{\mathbf{W}}$, the GPR rule can be rewritten as $\mathbf{Z}\mathbf{W} = \frac{1}{K+1}\left[\mathbf{X}, \mathbf{P}\mathbf{X}, \mathbf{P}^2\mathbf{X}, \cdots, \mathbf{P}^K\mathbf{X}\right]\mathbf{W}$. This constitutes a concatenation of the steps from 0 up to $K$. The learnable weight matrix $\mathbf{W} \in \mathbb{R}^{(K+1)F \times C}$ combines $\theta_k$ and $\bar{\mathbf{W}}$. These represent linearizations of GPR-GNNs (Chien et al., 2021b) and SIGNs (Frasca et al., 2020), simple yet useful models for learning on graphs. For simplicity, we only describe the results for node feature unlearning and delegate the analysis of edge and node unlearning to Appendix A.10.

**Theorem 4.6.** *Suppose that Assumption 4.2 holds and considers the node feature unlearning case. For $\mathbf{Z} = \frac{1}{K+1}\left[\mathbf{X}, \mathbf{P}\mathbf{X}, \mathbf{P}^2\mathbf{X}, \cdots, \mathbf{P}^K\mathbf{X}\right]$ and $\mathbf{P} = \tilde{\mathbf{D}}^{-1}\tilde{\mathbf{A}}$, we have*

$$\|\nabla L(\mathbf{w}^-, \mathcal{D}')\| = \|(\mathbf{H}_{\mathbf{w}_\eta} - \mathbf{H}_{\mathbf{w}^\star})\mathbf{H}_{\mathbf{w}^\star}^{-1}\Delta\| \leq \frac{\gamma_2(2c\lambda + (c\gamma_1 + \lambda c_1)\tilde{\mathbf{D}}_{mm})^2}{\lambda^4(m-1)}. \quad (6)$$

Note that the resulting bound is the same as the bound in Theorem 4.3. This is due to the fact that we used the normalization factor $\frac{1}{K+1}$ in $\mathbf{Z}$. Hence, given the same noise level, the GPR-based models are more sensitive when we trained on the noisy loss $L_\mathbf{b}$. Whether the general high-level performance of GPR can overcompensate this drawback depends on the actual datasets considered.

## 5 EMPIRICAL ASPECTS OF APPROXIMATE GRAPH UNLEARNING

**Logistic and least-squares regression on graphs.** For binary logistic regression, the loss equals $\ell(\mathbf{e}_i^T\mathbf{Z}\mathbf{w}, \mathbf{e}_i^T\mathbf{Y}_{T_r}) = -\log(\sigma(\mathbf{e}_i^T\mathbf{Y}_{T_r}\mathbf{e}_i^T\mathbf{Z}\mathbf{w}))$, where $\sigma(x) = 1/(1+\exp(-x))$ denotes the sigmoid function. As shown in Guo et al. (2020), the assumptions (1)-(3) in Assumption 4.2 are satisfied with $c = 1$ and $\gamma_2 = 1/4$. By standard analysis, we show that our loss satisfies (4) and (5) in Assumption 4.2 with $\gamma_1 = 1/4$ and $c_1 = 1$. For multi-class logistic regression, one can adapt the "one-versus-all other-classes" strategy which leads to the same result. For least-square regression, since the hessian is independent of $\mathbf{w}$ our approach offers $(0, 0)$-approximate graph unlearning even without loss perturbations. See Appendix A.5 for the complete discussion and derivation.

**Sequential unlearning.** In practice, multiple users may request unlearning. Hence, it is desirable to have a model that supports sequential unlearning of all types of data points. One can leverage the same proof as in Guo et al. (2020) (induction coupled with the triangle inequality) to show that the resulting gradient residual norm bound equals $T\epsilon'$ at the $T^{th}$ unlearning request, where $\epsilon'$ is the bound for a single instance of approximate graph unlearning.

**Data-dependent bounds.** The gradient residual norm bounds derived for different types of approximate graph unlearning contain a constant factor $1/\lambda^4$, and may be loose in practice. Following Guo et al. (2020), we also examined data dependent bounds.

**Corollary 5.1** (Application of Corollary 1 in Guo et al. (2020)). *For all three graph unlearning scenarios, we have $\|\nabla L(\mathbf{w}^-, \mathcal{D}')\| \leq \gamma_2\|\mathbf{Z}'\|_{op}\|\mathbf{H}_{\mathbf{w}^\star}^{-1}\Delta\|\|\mathbf{Z}'\mathbf{H}_{\mathbf{w}^\star}^{-1}\Delta\|$.*

Hence, there are two ways to accomplish approximate graph unlearning. If we do not allow any retraining, we have to leverage the worst case bound in Section 4 based on the expected number of unlearning requests. Importantly, we will also need to constrain the node degree of nodes to be unlearned (i.e., do not allow for unlearning hub nodes), for both node feature and node unlearning. Otherwise, we can select the noise standard deviation $\alpha$, $\epsilon$ and $\delta$ and compute the corresponding "privacy budget" $\alpha\epsilon/\sqrt{2\log(1.5/\delta)}$. Once the accumulated gradient residual norm exceeds this budget, we retrain the model from scratch. Note that this still greatly reduces the time complexity compare to retraining the model for every unlearning request (see Section 6). We also relegate the pseudo-code of our method leveraging data-dependent bounds for sequential unlearning in Appendix A.6.

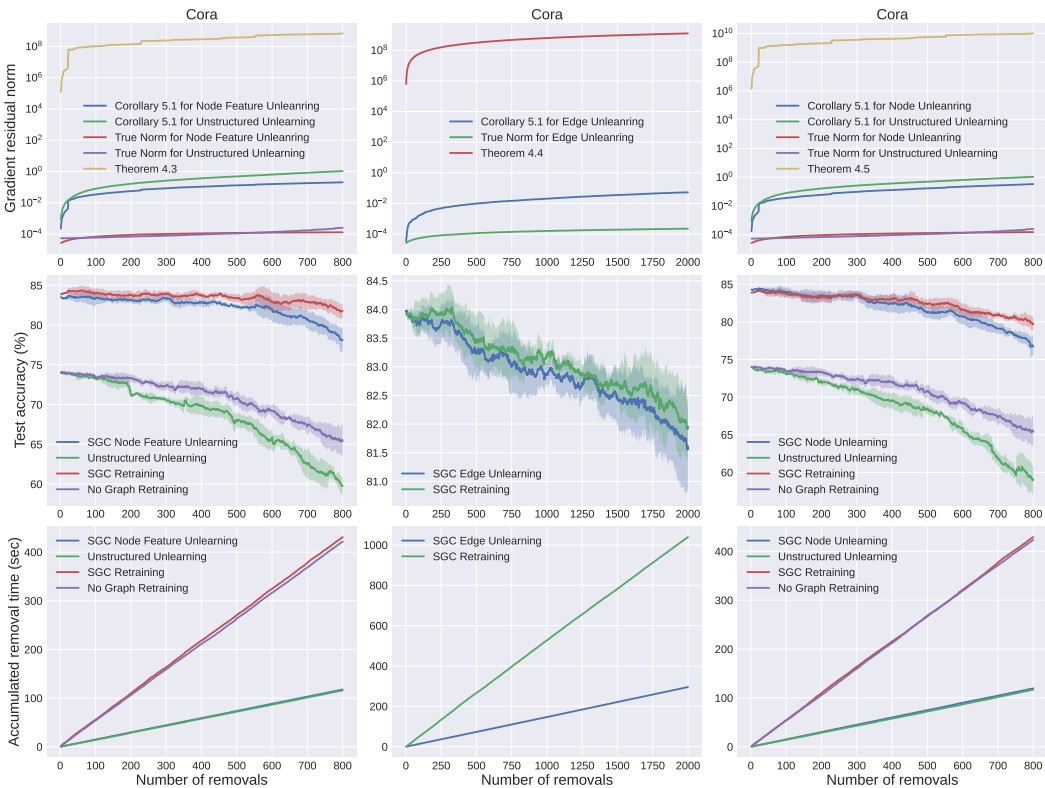

Figure 2: Comparison of proposed SGC node feature unlearning (left column), edge unlearning (middle column) and node unlearning (right column) with baseline methods. The shaded regions in the second row represent the standard deviation of test accuracy. In the third row, we show the accumulated unlearning time as a function of the number of unlearned points. The time needed for each unlearning procedure is given in Appendix A.19.

## 6 EXPERIMENT

**Settings.** We test our methods on benchmark datasets for graph learning, including Cora, Citseer, Pubmed (Sen et al., 2008; Yang et al., 2016; Fey & Lenssen, 2019) and large-scale dataset ogbn-arxiv (Hu et al., 2020) and Amazon co-purchase networks Computers and Photo (McAuley et al., 2015; Shchur et al., 2018). We either use the public splitting or random splitting based on similar rules as public splitting and focus on node classification. Following Guo et al. (2020), we use LBFGS as the optimizer for all methods due to its high efficiency on strongly convex problems. Unless specified otherwise, we fix $K = 2, \delta = 10^{-4}, \lambda = 10^{-2}, \epsilon = 1, \alpha = 0.1$ for all experiments, and average the results over 5 independent trails with random initializations. Our baseline methods include complete retraining with graph information after each unlearning request (SGC Retraining), complete retraining without graph information after each unlearning request (No Graph Retraining), and Unstructured Unlearning (Guo et al., 2020). Additional details can be found in Appendix A.19.

**Bounds on the gradient residual norm.** The first row of Figure 2 compares the values of both *worst-case* bounds computed in Section 4 and *data-dependent* bounds computed from Corollary 5.1 with the true value of the gradient residual norm (True Norm). For simplicity, we set $\alpha = 0$ during training. The observation is that the worst-case bounds are looser than the data-dependent bounds, and both bounds are indeed valid upper bounds for the actual gradient residual norm.

**Dependency on node degrees.** While an upper bound does not necessarily capture the dependency of each term correctly, we show in Figure 3 (a) and (b) that our Theorem 4.5 and 4.6 indeed do so. Here, each point corresponds to unlearning one node. We test for all nodes in the training set $T_r$ and fix $\lambda = 10^{-4}, \alpha = 0$. Our results show that unlearning a large-degree node is more expensive in terms of the privacy budget (i.e., it induces a larger gradient residual norm). For other datasets, refer to Appendix A.19.

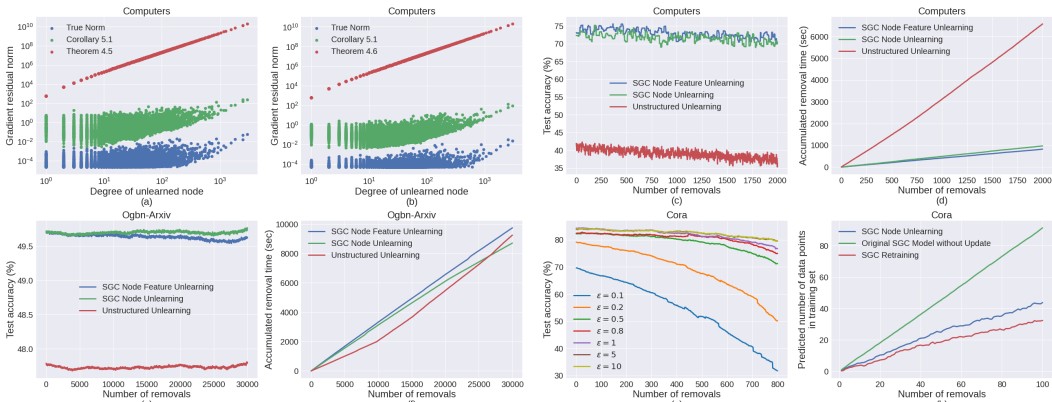

Figure 3: (a), (b) Simulation verification of the result in Theorem 4.5 and 4.6 pertaining to node degrees. (d), (f) Accumulated unlearning time as a function of the number of removed nodes. The unlearning time of Unstructured Unlearning is often higher than that of our proposed approximate graph unlearning algorithms, because the number of retraining steps needed may be larger. (c), (e) Performance of approximate graph unlearning methods on different datasets. We set $\alpha = 10, \lambda = 10^{-4}$ for Computers and $\lambda = 10^{-4}$ for ogbn-arxiv. The number of repeated trials is 3 due to the large amount of removed data. (g) Tradeoff between privacy $\epsilon$ and performance. To achieve similar numbers of retraining, we set $\alpha\epsilon = 0.1$. (h) The number of data points predicted by the membership inference attack model to lie in the training set.

**Performance of approximate graph unlearning methods.** The time complexity of unlearning and test accuracy of our proposed approximate graph unlearning methods after unlearning is shown in Figures 2, and 3. It shows that: (1) Leveraging graph information is necessary when designing unlearning methods for node classification tasks. (2) Our method supports unlearning a large proportion of data points with a small loss in test accuracy. (3) Our method is around $4\times$ faster than completely retraining the model after each unlearning request. (4) Our methods have robust performance regardless of the scale of the datasets. For more results see Appendix A.19.

**Trade-off amongst privacy, performance and time complexity.** As indicated in Theorem 4.1, there is a trade-off amongst privacy, performance and time complexity. Comparing to exact unlearning (i.e. SGC retraining), allowing approximate unlearning gives $4\times$ speedup in time with competitive performance. We further examine this trade-off by fixing $\lambda$ and $\delta$, then the trade-off is controlled by $\epsilon$ and $\alpha$. The results are shown in Figure 3 (g) for Cora, where we set $\alpha\epsilon = 0.1$. The test accuracy increases when we relax our constraints on $\epsilon$, which agrees with our intuition. Remarkably, we can still obtain competitive performance with SGC Retraining when we require $\epsilon$ to be as small as 1. In contrast, one needs at least $\epsilon \geq 5$ to unlearn even *one* node or edge by leveraging state-of-the-art DP-GNNs (Sajadmanesh et al., 2022; Daigavane et al., 2021) for reasonable performance, albeit our tested datasets are different. This shows the benefit of our approximate graph unlearning method as opposed to both retraining from scratch and DP-GNNs. Unfortunately, the codes of these DP-GNNs are not publicly available, which prevents us from testing them on our datasets in a unified treatment.

**Membership inference attacks on unlearned models.** We performed experiments pertaining to the node unlearning task and applied the membership inference (MI) attack for GNNs reported in Olatunji et al. (2021b) on our updated model. The experimental details are discussed in Appendix A.19. As shown in Figure 3 (h), even for full *SGC retraining* the attack can still identify parts of the removed nodes in the training set (for relevant explanations, see Appendix A.2), and the result of *SGC node unlearning* is slightly worse (w.r.t privacy) than retraining since our algorithm performs approximate unlearning. Note that the performance of the MI attack on the original model is consistent with the results in Olatunji et al. (2021b) and significantly worse than both our unlearning and complete retraining method. The results also highlight the fact that the privacy definition considered in MI attacks and approximate unlearning are different (see Appendix A.2). Nevertheless, experiments show that our method offers similar privacy-preserving performance (in terms of MI) as complete retraining, and better performance compared to just using the original model without unlearning features.

ACKNOWLEDGMENTS

This work was funded by NSF grants 1816913 and 1956384. The authors thank Wei-Ning Chen and Pan Li for the helpful discussion.

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

## A  APPENDIX

### A.1  CONCLUSION

We introduced the first known framework for approximate graph unlearning. In this setting, new analytical unlearning challenges had to be addressed due to the presence of complex graph feature and topology data. Our analytical contributions pertain to novel proof techniques for approximate graph unlearning, while our empirical studies on six benchmark datasets established fundamental performance-complexity trade-offs between unlearning and complete retraining.

### A.2 FUTURE RESEARCH DIRECTIONS AND LIMITATIONS

**Batch unlearning.** In practice, it is likely that we not only require sequential unlearning, but also batch unlearning: A number of users may request their data to be unlearned within a certain (short) time frame. The approach in Guo et al. (2020) can ensure certified removal even in this scenario. The generalization of our approach for batch unlearning is also possible, but will be discussed elsewhere.

**Nonlinear models and hypergraph extension.** Also akin to what was described in Guo et al. (2020), we can leverage pre-trained (nonlinear) feature extractors or special graph feature transforms to further improve the performance of the overall model. For example, Chien et al. (2022a) proposed a node feature extraction method termed GIANT-XRT that greatly improves the performance of simple network models such as MLP and SGC. If a public dataset is never subjected to unlearning, one can pre-train GIANT-XRT on that dataset and use it for subsequent approximate graph unlearning. If such a public dataset is unavailable, we have to make the node feature extractor DP. In this case, we can either design a DP version of GIANT-XRT or leverage the DP-GNN model described in Section 2. By applying Theorem 5 of Guo et al. (2020), the overall model can be shown to guarantee approximate graph unlearning, where the parameters $\epsilon$ and $\delta$ now also depend on the DP guarantees of the node feature extractor. There is also another line of work on Graph Scattering Transforms (GSTs) (Gama et al., 2019; Pan et al., 2021) for use as feature extractors for graph information. Since a GST is a predefined mathematical transform and hence does not require training, it can be easily combined with our approach (Pan et al., 2022). Finally, generalizing approximate graph unlearning to hypergraphs can also be an interesting direction. Although the current SOTA hypergraph neural networks heavily rely on nonlinear modules such as AllSet (Chien et al., 2022b), we believe extension to classical hypergraph learning algorithms (Chien et al., 2021a) is possible.

**Empirical metrics and MI.** There is currently no empirical metric that can be used to evaluate how well *approximate* machine unlearning methods preserve privacy. Although the definition of approximate graph unlearning automatically and theoretically ensures that one cannot infer information about the unlearned data point from the updated model (if one chooses to set $\epsilon, \delta$ to 0), it remains an open problem whether we can design an empirical metric that can accurately quantify this privacy-preserving performance. Note that privacy-based attacks like the membership inference attack (Shokri et al., 2017; Olatunji et al., 2021b) have completely different design goals and may not work well in unlearning practice. For example, assume that there are two nodes that share similar features and neighborhood structures, come from the same class in the graph and are both included in the training set. This scenario frequently arises in practice, especially for graphs with strong homophily properties. In this case, even if we unlearn one of the nodes, the attack model will still have a high probability of recognizing the unlearned node in the training set due to the presence of the "similar" node. Thus, the viability of using the results returned by the attack models to assess the performance of an unlearner is not clear. This is also verified by some preliminary experiments on node unlearning tasks described in Section A.19 and the main text.

**Societal impacts.** The authors believe that for medical and biological sciences research, the right to be forgotten may significantly set back potentially life-saving discoveries due to the need to have access to many diverse data samples. But current trends seem to favor privacy over discovery rates and timings. Hence, a compromise between data availability and the right to be forgotten has to be established in the near future.

One current limitation of our work is that the newly proposed proof techniques do not apply to general graph neural networks where nonlinear activation functions are used. Nevertheless, our work is the first step towards developing approximate graph unlearning approaches for general GNNs.

### A.3 INTUITION BEHIND THE MODEL UPDATE RULE

Our unlearning mechanism proposed in Section 4 is

$$\mathbf{w}^- = \mathbf{w}^\star + \left[\nabla^2 L(\mathbf{w}^\star, \mathcal{D}')\right]^{-1} \left[\nabla L(\mathbf{w}^\star, \mathcal{D}) - \nabla L(\mathbf{w}^\star, \mathcal{D}')\right],$$

and the intuition is stated as follows. Our goal for the updated model is $\nabla L(\mathbf{w}^-, \mathcal{D}') = 0$. By Taylor series we have that

$$\nabla L(\mathbf{w}^-, \mathcal{D}') \approx \nabla L(\mathbf{w}^\star, \mathcal{D}') + \nabla^2 L(\mathbf{w}^\star, \mathcal{D}')(\mathbf{w}^- - \mathbf{w}^\star) = 0.$$

Therefore, we have

$$\mathbf{w}^- - \mathbf{w}^\star = \left[\nabla^2 L(\mathbf{w}^\star, \mathcal{D}')\right]^{-1} \left[0 - \nabla L(\mathbf{w}^\star, \mathcal{D}')\right]$$

$$\mathbf{w}^- = \mathbf{w}^\star + \left[\nabla^2 L(\mathbf{w}^\star, \mathcal{D}')\right]^{-1} \left[\nabla L(\mathbf{w}^\star, \mathcal{D}) - \nabla L(\mathbf{w}^\star, \mathcal{D}')\right].$$

The last equality holds due to the fact that $\mathbf{w}^\star$ should be the unique optimizer for the strongly convex loss $L(\mathbf{w}, \mathcal{D})$ over the entire dataset $\mathcal{D}$.

## A.4 ADDITIONAL ILLUSTRATIONS

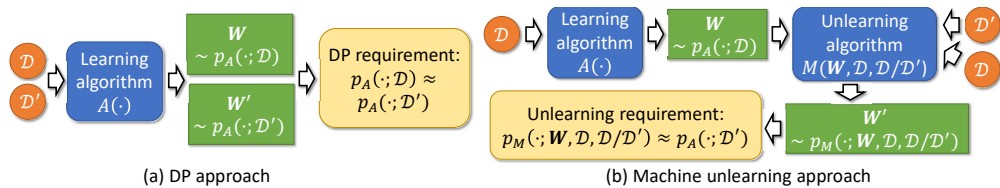

(a) DP approach                (b) Machine unlearning approach

Figure 4: Difference between machine unlearning (as defined in Guo et al. (2020)) and Differential Privacy (DP).

## A.5 ADDITIONAL DISCUSSIONS

**Details on on Assumption 4.2.** Assumptions (2), (4) and (5) in our model and that of Guo et al. (2020) require Lipschitz conditions with respect to the first argument of $\ell$, but not the second. We also implicitly assume that the second argument (corresponding to labels) does not effect the norm of gradients or Hessians. One example that meets these constraints is the logistic loss: If $\ell(\mathbf{w}^T\mathbf{x}, y) = \ell(y\mathbf{w}^T\mathbf{x})$ then all required assumptions hold.

**Least-squares and logistic regression on graphs.** Paralleling once again the results of Guo et al. (2020), it is clear that our approximate graph unlearning mechanism can be used in conjunction with least-squares and logistic regressions. For example, node classification can be performed using a logistic loss. The node regression problem described in Ma et al. (2020); Jia & Benson (2020) is related to least-squares regression. In particular, least-squares regression uses the loss $\ell(\mathbf{e}_i^T \mathbf{Z}\mathbf{w}, \mathbf{e}_i^T \mathbf{Y}_{T_r}) = (\mathbf{e}_i^T \mathbf{Z}\mathbf{w} - \mathbf{e}_i^T \mathbf{Y}_{T_r})^2$. Note that its Hessian is of the form $(\mathbf{e}_i^T \mathbf{Z})^T \mathbf{e}_i^T \mathbf{Z}$, which does not depend on $\mathbf{w}$. Thus, based on the same arguments presented in Guo et al. (2020), our proposed unlearning method $M$ offers $(0,0)$-approximate graph unlearning even without loss perturbations.

For binary logistic regression, the loss equals $\ell(\mathbf{e}_i^T \mathbf{Z}\mathbf{w}, \mathbf{e}_i^T \mathbf{Y}_{T_r}) = -\log(\sigma(\mathbf{e}_i^T \mathbf{Y}_{T_r} \mathbf{e}_i^T \mathbf{Z}\mathbf{w}))$, where $\sigma(x) = 1/(1 + \exp(-x))$ denotes the sigmoid function. As shown in Guo et al. (2020), the assumptions (1)-(3) in 4.2 are satisfied with $c = 1$ and $\gamma_2 = 1/4$. We only need to show that (4) and (5) of 4.2 hold as well. Observe that $\ell'(x, \mathbf{e}_i^T \mathbf{Y}_{T_r}) = \left(\sigma(\mathbf{e}_i^T \mathbf{Y}_{T_r} x) - 1\right)$. Since the sigmoid function $\sigma(\cdot)$ is restricted to lie in $[0,1]$, $|\ell'|$ is bounded by 1, which means that our loss satisfies (5) in 4.2 with $c_1 = 1$. Based on the Mean Value Theorem, one can show that $\sigma(x)$ is $\max_{x \in \mathbb{R}} |\sigma(x)'|$-Lipschitz. Using some simple algebra, one can also prove that $\sigma(x)' = \sigma(x)(1 - \sigma(x)) \Rightarrow \max_{x \in \mathbb{R}} |\sigma(x)'| = 1/4$. Thus our loss satisfies assumption (4) in 4.2 as well, with $\gamma_1 = 1/4$. For multi-class logistic regression, one can adapt the "one-versus-all other-classes" strategy which leads to the same result.

## A.6 ALGORITHMIC DETAILS

The pseudo-codes for training removal-enabled models and the removal procedure for the case of binary classification are presented below. Note that this procedure is the same for all three types of removal requests (node feature unlearning, edge unlearning and node unlearning). During training, we add a random linear term to the training loss by sampling a Gaussian noise vector $\mathbf{b}$. The choice of standard deviation $\alpha$ determines the "privacy budget" $\alpha\epsilon/\sqrt{2\log(1.5/\delta)}$ as shown in Section 5.

---

**Algorithm 1** Training procedure

---

1: **input:** Training data $\mathbf{Z} \in \mathbb{R}^{m \times d}$, training labels $\mathbf{Y} \in \mathbb{R}^m$, loss $\ell$, parameters $\alpha, \lambda > 0$.
2: Sample the noise vector $\mathbf{b} \sim \mathcal{N}(0, \alpha^2)^d$.
3: $\mathbf{w}^\star = \arg\min_{\mathbf{w} \in \mathbb{R}^d} \sum_{i=1}^m \left( \ell(\mathbf{z}_i^T \mathbf{w}, y_i) + \frac{\lambda}{2} \|\mathbf{w}\|^2 \right) + \mathbf{b}^T \mathbf{w}$.
4: **return** $\mathbf{w}^\star$.

---

**Algorithm 2** Unlearning procedure

---

1: **input:** Feature matrix $\mathbf{X} \in \mathbb{R}^{n \times d}$, labels $\mathbf{Y} \in \mathbb{R}^n$, one-step propagation matrix $\mathbf{P}$, loss $\ell$, training set indices $T_r = \{i_1, i_2, \ldots\}$, sequence of removal requests $R_m = \{j_1, j_2, \ldots\}$, parameters $K, \epsilon, \delta, \gamma_2, \alpha, \lambda > 0$.
2: Compute node embedding after propagation $\mathbf{Z} = \mathbf{P}^K \mathbf{X}$.
3: Training set $\mathcal{D} = \{\mathbf{z}_i, y_i\}_{i \in T_r}$.
4: Compute $\mathbf{w}$ using Algorithm 1 $(\mathcal{D}, \ell, \alpha, \lambda)$.
5: Accumulated gradient residual norm $\beta = 0$.
6: **for** $j \in R_m$ **do**
7:     Update the feature matrix $\mathbf{X}'$ and propagation matrix $\mathbf{P}'$ based on the removal.
8:     Compute new node embedding after propagation $\mathbf{Z}' = \mathbf{P}'^K \mathbf{X}'$.
9:     **if** $j \in Tr$ **then**
10:         Remove $j$ from the training indices $T_r = T_r \setminus \{j\}$.
11:     **end if**
12:     Update the training set $\mathcal{D}' = \{\mathbf{z}'_i, y_i\}_{i \in T_r}$.
13:     Compute $\Delta = \nabla L(\mathbf{w}, \mathcal{D}) - \nabla L(\mathbf{w}, \mathcal{D}')$.
14:     Compute $\mathbf{H} = \nabla^2 L(\mathbf{w}; \mathcal{D}')$.
15:     Update accumulated gradient residual norm $\beta = \beta + \gamma_2 \|\mathbf{Z}'\|_{op} \|\mathbf{H}^{-1}\Delta\| \|\mathbf{Z}'\mathbf{H}^{-1}\Delta\|$.
16:     **if** $\beta > \alpha\epsilon / \sqrt{2\log(1.5/\delta)}$ **then**
17:         Recompute $\mathbf{w}$ using Algorithm 1 $(\mathcal{D}', \ell, \alpha, \lambda)$, $\beta = 0$.
18:     **else**
19:         $\mathbf{w} = \mathbf{w} + \mathbf{H}^{-1}\Delta$.
20:     **end if**
21: **end for**
22: **return** $\mathbf{w}$.

---

### A.7 PROOF OF THEOREM 4.3

**Theorem.** *Under the node feature unlearning scenario, $\mathcal{D} = (\mathbf{X}, \mathbf{Y}_{T_r}, \mathbf{A})$ and $\mathcal{D} = (\mathbf{X}', \mathbf{Y}'_{T_r}, \mathbf{A})$. Suppose Assumption 4.2 holds. For $\mathbf{Z} = \mathbf{P}^K \mathbf{X}$ and $\mathbf{P} = \tilde{\mathbf{D}}^{-1}\tilde{\mathbf{A}}$, we have*

$$\|\nabla L(\mathbf{w}^-, \mathcal{D}')\| = \|(\mathbf{H}_{\mathbf{w}_\eta} - \mathbf{H}_{\mathbf{w}^\star})\mathbf{H}_{\mathbf{w}^\star}^{-1}\Delta\| \leq \frac{\gamma_2(2c\lambda + (c\gamma_1 + \lambda c_1)\tilde{\mathbf{D}}_{mm})^2}{\lambda^4(m-1)}. \tag{7}$$

We need to ensure that the norm of each row of $\mathbf{Z}$ is bounded by 1. We state the following lemma in support of this claim.

**Lemma A.1.** *Assume that $\|\mathbf{e}_i^T \mathbf{S}\| \leq 1$, $\forall i \in [n]$. Then, $\forall i \in [n], K \geq 0$, $\|\mathbf{e}_i^T \mathbf{P}^K \mathbf{S}\| \leq 1$, where $\mathbf{P} = \tilde{\mathbf{D}}^{-1}\tilde{\mathbf{A}}$.*

*Proof.* Our proof is a nontrivial generalization and extension of the proof in Guo et al. (2020). For completeness, we outline every step of the proof. We also emphasize novel approaches used to accommodate out approximate graph unlearning scenario.

Let $G(\mathbf{w}) = \nabla L(\mathbf{w}, \mathcal{D}')$. By the Taylor theorem, $\exists \eta \in [0, 1]$ such that

$$G(\mathbf{w}^-) = G(\mathbf{w}^\star + \mathbf{H}_{\mathbf{w}^\star}^{-1}\Delta) = G(\mathbf{w}^\star) + \nabla G(\mathbf{w}^\star + \eta\mathbf{H}_{\mathbf{w}^\star}^{-1}\Delta)\mathbf{H}_{\mathbf{w}^\star}^{-1}\Delta$$

$$\overset{(a)}{=} G(\mathbf{w}^\star) + \mathbf{H}_{\mathbf{w}_\eta}\mathbf{H}_{\mathbf{w}^\star}^{-1}\Delta$$

$$= G(\mathbf{w}^\star) + \Delta + \mathbf{H}_{\mathbf{w}_\eta}\mathbf{H}_{\mathbf{w}^\star}^{-1}\Delta - \Delta$$

$$\overset{(b)}{=} 0 + \mathbf{H}_{\mathbf{w}_\eta}\mathbf{H}_{\mathbf{w}^\star}^{-1}\Delta - \Delta$$

$$= \mathbf{H}_{\mathbf{w}_\eta}\mathbf{H}_{\mathbf{w}^\star}^{-1}\Delta - \mathbf{H}_{\mathbf{w}^\star}\mathbf{H}_{\mathbf{w}^\star}^{-1}\Delta$$

$$= (\mathbf{H}_{\mathbf{w}_\eta} - \mathbf{H}_{\mathbf{w}^\star})\mathbf{H}_{\mathbf{w}^\star}^{-1}\Delta. \tag{8}$$

In (a), we wrote $\mathbf{H}_{\mathbf{w}_\eta} \triangleq \nabla G(\mathbf{w}^\star + \eta\mathbf{H}_{\mathbf{w}^\star}^{-1}\Delta)$, corresponding to the Hessian at $\mathbf{w}_\eta \triangleq \mathbf{w}^\star + \eta\mathbf{H}_{\mathbf{w}^\star}^{-1}\Delta$. Equality (b) is due to our choice of $\Delta = \nabla L(\mathbf{w}^\star, \mathcal{D}) - \nabla L(\mathbf{w}^\star, \mathcal{D}')$ and the fact that $\mathbf{w}^\star$ is the minimizer of $L(\cdot, \mathcal{D})$. We would like to point out that our choice of $\Delta$ is more general then that Guo et al. (2020): Since unlearning one node may affect the entire node embedding $\mathbf{Z}$, a generalization of $\Delta$ is crucial. When $K = 0$ (i.e., when no graph topology is included), one recovers $\Delta$ from Guo et al. (2020) as a special case of our model. In the latter part of the proof, we will see how the graph setting makes the analysis more intricate and complex.

By the Cauchy-Schwartz inequality, we have

$$\|G(\mathbf{w}^-)\| \leq \|\mathbf{H}_{\mathbf{w}_\eta} - \mathbf{H}_{\mathbf{w}^\star}\| \|\mathbf{H}_{\mathbf{w}^\star}^{-1}\Delta\|. \tag{9}$$

Below we bound both norms on the right hand side separately. We start with the term $\|\mathbf{H}_{\mathbf{w}_\eta} - \mathbf{H}_{\mathbf{w}^\star}\|$. Note that

$$\|\nabla^2\ell(\mathbf{e}_i^T\mathbf{Z}'\mathbf{w}_\eta, \mathbf{e}_i^T\mathbf{Y}'_{T_r}) - \nabla^2\ell(\mathbf{e}_i^T\mathbf{Z}'\mathbf{w}_\star, \mathbf{e}_i^T\mathbf{Y}'_{T_r})\|$$

$$= \|\left[\ell''(\mathbf{e}_i^T\mathbf{Z}'\mathbf{w}_\eta, \mathbf{e}_i^T\mathbf{Y}'_{T_r}) - \ell''(\mathbf{e}_i^T\mathbf{Z}'\mathbf{w}_\star, \mathbf{e}_i^T\mathbf{Y}'_{T_r})\right](\mathbf{e}_i^T\mathbf{Z}')^T\mathbf{e}_i^T\mathbf{Z}'\|$$

$$\overset{(a)}{\leq} \gamma_2\|\mathbf{e}_i^T\mathbf{Z}'\mathbf{w}_\eta - \mathbf{e}_i^T\mathbf{Z}'\mathbf{w}^\star\|\|\mathbf{e}_i^T\mathbf{Z}'\|^2$$

$$\leq \gamma_2\|\mathbf{w}_\eta - \mathbf{w}^\star\|\|\mathbf{e}_i^T\mathbf{Z}'\|^3 = \gamma_2\|\eta\mathbf{H}_{\mathbf{w}^\star}^{-1}\Delta\|\|\mathbf{e}_i^T\mathbf{Z}'\|^3 \leq \gamma_2\|\mathbf{H}_{\mathbf{w}^\star}^{-1}\Delta\|\|\mathbf{e}_i^T\mathbf{Z}'\|^3. \tag{10}$$

Here, (a) follows from the Cauchy-Schwartz inequality and the Lipschitz condition on $\ell''$ in Assumption 4.2. Unlike the analysis in Guo et al. (2020), we are faced with the problem of bounding the term $\|\mathbf{e}_i^T\mathbf{Z}'\|$. In Guo et al. (2020) (where $\mathbf{Z} = \mathbf{X}$), a simple bound equals 1, which may be ontained via (3) in Assumption 4.2. However, in our case, due to graph propagation this norm needs more careful examination and a simple application of the Cauchy-Schwartz inequality does not suffice, as it would lead to a term $\|\mathbf{X}\|_{op}$, where $\|\cdot\|_{op}$ denotes the operator norm. The simple worst case (i.e., when all rows of $\mathbf{X}$ are identical) leads to a meaningless bound $O(n)$.

By leveraging Lemma A.1, we can further upper bound equation 10 according to

$$\|\nabla^2\ell(\mathbf{e}_i^T\mathbf{Z}'\mathbf{w}_\eta, \mathbf{e}_i^T\mathbf{Y}'_{T_r}) - \nabla^2\ell(\mathbf{e}_i^T\mathbf{Z}'\mathbf{w}_\star, \mathbf{e}_i^T\mathbf{Y}'_{T_r})\| \leq \gamma_2\|\mathbf{H}_{\mathbf{w}^\star}^{-1}\Delta\|\|\mathbf{e}_i^T\mathbf{Z}'\|^3$$

$$\overset{(a)}{\leq} \gamma_2\|\mathbf{H}_{\mathbf{w}^\star}^{-1}\Delta\|, \tag{11}$$

where (a) follows from Lemma A.1.

As a result, we arrive at a bound for $\|\mathbf{H}_{\mathbf{w}_\eta} - \mathbf{H}_{\mathbf{w}^\star}\|$ of the form

$$
\|\mathbf{H}_{\mathbf{w}_\eta} - \mathbf{H}_{\mathbf{w}^\star}\| \leq \sum_{i=1}^{m-1} \|\nabla^2 \ell(\mathbf{e}_i^T \mathbf{Z}' \mathbf{w}_\eta, \mathbf{e}_i^T \mathbf{Y}'_{T_r}) - \nabla^2 \ell(\mathbf{e}_i^T \mathbf{Z}' \mathbf{w}_\star, \mathbf{e}_i^T \mathbf{Y}'_{T_r})\|
$$
$$
\leq \gamma_2 (m-1) \|\mathbf{H}_{\mathbf{w}^\star}^{-1} \Delta\|. \tag{12}
$$

Next, we bound $\|\mathbf{H}_{\mathbf{w}^\star}^{-1} \Delta\|$. Since $L(\cdot, \mathcal{D}')$ is $\lambda(m-1)$-strongly convex, we have $\|\mathbf{H}_{\mathbf{w}^\star}^{-1}\| \leq \frac{1}{\lambda(m-1)}$. For the norm $\|\Delta\|$, we have

$$
\Delta = \nabla L(\mathbf{w}^\star, \mathcal{D}) - \nabla L(\mathbf{w}^\star, \mathcal{D}')
$$
$$
= \lambda \mathbf{w}^\star + \nabla \ell(\mathbf{e}_m^T \mathbf{Z} \mathbf{w}^\star, \mathbf{e}_m^T \mathbf{Y}_{T_r}) + \sum_{i=1}^{m-1} \left[ \nabla \ell(\mathbf{e}_i^T \mathbf{Z} \mathbf{w}^\star, \mathbf{e}_i^T \mathbf{Y}_{T_r}) - \nabla \ell(\mathbf{e}_i^T \mathbf{Z}' \mathbf{w}^\star, \mathbf{e}_i^T \mathbf{Y}_{T_r}) \right]. \tag{13}
$$

The third term does not appear in Guo et al. (2020), since when $K = 0$, $\mathbf{Z} = \mathbf{X}$ and $\mathbf{Z}' = \mathbf{X}'$ are identical except for the $m^{th}$ row. In the approximate graph unlearning scenario, even removing one node feature can make the entire node embedding matrix $\mathbf{Z}$ change in every row, which creates new analytical challenges. For example, consider the case $\mathbf{X} = [x_1, x_2, x_3]^T$, where we have a graph with three nodes, each with a 1-dimensional feature. Consider the fully connected graph (i.e., all entries in $\mathbf{P}$ set to $1/3$). Then, unlearning node 1 results in $\mathbf{Z}' = [0, x_2, x_3]^T$ for unstructured unlearning. However, $\mathbf{Z}' = [(x_2+x_3)/3, (x_2+x_3)/3, (x_2+x_3)/3]^T$ for the case of $L = 1$, which is completely different from $\mathbf{Z} = [(x_1+x_2+x_3)/3, (x_1+x_2+x_3)/3, (x_1+x_2+x_3)/3]^T$. Hence, the analysis in Guo et al. (2020) cannot be directly applied to graphs, as $\mathbf{Z}'$ changes in more than just one row compared to $\mathbf{Z}$ while unlearning a node feature.

By Minkowski's triangle inequality, we only need to bound the norm of the three individual terms in order to bound the norm of $\Delta$. For $\|\mathbf{w}^\star\|$, since $\mathbf{w}^\star$ is the global optimum of $L(\cdot; \mathcal{D})$, we have

$$
0 = \nabla L(\mathbf{w}^\star; \mathcal{D}) = \sum_{i=1}^{m} \nabla \ell(\mathbf{e}_i^T \mathbf{Z} \mathbf{w}^\star, \mathbf{e}_i^T \mathbf{Y}_{T_r}) + \lambda m \mathbf{w}^\star. \tag{14}
$$

By (1) in Assumption 4.2, we have

$$
\|\mathbf{w}^\star\| = \frac{\|\sum_{i=1}^{m} \nabla \ell(\mathbf{e}_i^T \mathbf{Z} \mathbf{w}^\star, \mathbf{e}_i^T \mathbf{Y}_{T_r})\|}{\lambda m} \leq \frac{c}{\lambda}. \tag{15}
$$

Once again, by (1) in Assumption 4.2, we have

$$
\|\nabla \ell(\mathbf{e}_m^T \mathbf{Z} \mathbf{w}^\star, \mathbf{e}_m^T \mathbf{Y}_{T_r})\| \leq c. \tag{16}
$$

A bound for the last term is established in the last step, as described below.

$$
\| \sum_{i=1}^{m-1} \left[ \nabla \ell(\mathbf{e}_i^T \mathbf{Z} \mathbf{w}^\star, \mathbf{e}_i^T \mathbf{Y}_{T_r}) - \nabla \ell(\mathbf{e}_i^T \mathbf{Z}' \mathbf{w}^\star, \mathbf{e}_i^T \mathbf{Y}_{T_r}) \right] \|
$$
$$
\leq \sum_{i=1}^{m-1} \| \nabla \ell(\mathbf{e}_i^T \mathbf{Z} \mathbf{w}^\star, \mathbf{e}_i^T \mathbf{Y}_{T_r}) - \nabla \ell(\mathbf{e}_i^T \mathbf{Z}' \mathbf{w}^\star, \mathbf{e}_i^T \mathbf{Y}_{T_r}) \|
$$
$$
= \sum_{i=1}^{m-1} \| \ell'(\mathbf{e}_i^T \mathbf{Z} \mathbf{w}^\star, \mathbf{e}_i^T \mathbf{Y}_{T_r})(\mathbf{e}_i^T \mathbf{Z})^T - \ell'(\mathbf{e}_i^T \mathbf{Z}' \mathbf{w}^\star, \mathbf{e}_i^T \mathbf{Y}_{T_r})(\mathbf{e}_i^T \mathbf{Z}')^T \|. \tag{17}
$$

Observe that

$$
\| \ell'(\mathbf{e}_i^T \mathbf{Z} \mathbf{w}^\star, \mathbf{e}_i^T \mathbf{Y}_{T_r})(\mathbf{e}_i^T \mathbf{Z})^T - \ell'(\mathbf{e}_i^T \mathbf{Z}' \mathbf{w}^\star, \mathbf{e}_i^T \mathbf{Y}_{T_r})(\mathbf{e}_i^T \mathbf{Z}')^T \|
$$
$$
\leq \| \ell'(\mathbf{e}_i^T \mathbf{Z} \mathbf{w}^\star, \mathbf{e}_i^T \mathbf{Y}_{T_r})(\mathbf{e}_i^T \mathbf{Z})^T - \ell'(\mathbf{e}_i^T \mathbf{Z}' \mathbf{w}^\star, \mathbf{e}_i^T \mathbf{Y}_{T_r})(\mathbf{e}_i^T \mathbf{Z})^T \|
$$
$$
+ \| \ell'(\mathbf{e}_i^T \mathbf{Z}' \mathbf{w}^\star, \mathbf{e}_i^T \mathbf{Y}_{T_r})(\mathbf{e}_i^T \mathbf{Z})^T - \ell'(\mathbf{e}_i^T \mathbf{Z}' \mathbf{w}^\star, \mathbf{e}_i^T \mathbf{Y}_{T_r})(\mathbf{e}_i^T \mathbf{Z}')^T \| \tag{18}
$$

The first term can be bounded as

$$
\begin{aligned}
& \|\ell'(\mathbf{e}_i^T \mathbf{Z} \mathbf{w}^\star, \mathbf{e}_i^T \mathbf{Y}_{T_r})(\mathbf{e}_i^T \mathbf{Z})^T - \ell'(\mathbf{e}_i^T \mathbf{Z}' \mathbf{w}^\star, \mathbf{e}_i^T \mathbf{Y}_{T_r})(\mathbf{e}_i^T \mathbf{Z})^T\| \\
& \leq \left|\ell'(\mathbf{e}_i^T \mathbf{Z} \mathbf{w}^\star, \mathbf{e}_i^T \mathbf{Y}_{T_r}) - \ell'(\mathbf{e}_i^T \mathbf{Z}' \mathbf{w}^\star, \mathbf{e}_i^T \mathbf{Y}_{T_r})\right| \|(\mathbf{e}_i^T \mathbf{Z})^T\| \\
& \overset{(a)}{\leq} \gamma_1 \|\mathbf{e}_i^T \mathbf{Z} \mathbf{w}^\star - \mathbf{e}_i^T \mathbf{Z}' \mathbf{w}^\star\| \|(\mathbf{e}_i^T \mathbf{Z})^T\| \\
& \overset{(b)}{\leq} \gamma_1 \|(\mathbf{e}_i^T \mathbf{Z} - \mathbf{e}_i^T \mathbf{Z}')^T\| \|\mathbf{w}^\star\| \\
& \overset{(c)}{\leq} \frac{c \gamma_1}{\lambda} \|(\mathbf{e}_i^T \mathbf{Z} - \mathbf{e}_i^T \mathbf{Z}')^T\|.
\end{aligned}
\tag{19}
$$

Here, (a) is due to (4) in Assumption 4.2, while (b) follows from Lemma A.1 and the Cauchy-Schwartz inequality. Inequality (c) is a consequence of the bound for $\|\mathbf{w}\|$ that we previously derived.

The second term can be bounded as

$$
\begin{aligned}
& \|\ell'(\mathbf{e}_i^T \mathbf{Z}' \mathbf{w}^\star, \mathbf{e}_i^T \mathbf{Y}_{T_r})(\mathbf{e}_i^T \mathbf{Z})^T - \ell'(\mathbf{e}_i^T \mathbf{Z}' \mathbf{w}^\star, \mathbf{e}_i^T \mathbf{Y}_{T_r})(\mathbf{e}_i^T \mathbf{Z}')^T\| \\
& \leq \left|\ell'(\mathbf{e}_i^T \mathbf{Z}' \mathbf{w}^\star, \mathbf{e}_i^T \mathbf{Y}_{T_r})\right| \|(\mathbf{e}_i^T \mathbf{Z})^T - (\mathbf{e}_i^T \mathbf{Z}')^T\| \\
& \overset{(a)}{\leq} c_1 \|(\mathbf{e}_i^T \mathbf{Z})^T - (\mathbf{e}_i^T \mathbf{Z}')^T\|.
\end{aligned}
\tag{20}
$$

For the inequality in (a), we used (5) from Assumption 4.2. Put together, we have

$$
\begin{aligned}
& \left\| \sum_{i=1}^{m-1} \left[ \nabla \ell(\mathbf{e}_i^T \mathbf{Z} \mathbf{w}^\star, \mathbf{e}_i^T \mathbf{Y}_{T_r}) - \nabla \ell(\mathbf{e}_i^T \mathbf{Z}' \mathbf{w}^\star, \mathbf{e}_i^T \mathbf{Y}_{T_r}) \right] \right\| \\
& \leq \sum_{i=1}^{m-1} \left[ \left( \frac{c \gamma_1}{\lambda} + c_1 \right) \|(\mathbf{e}_i^T \mathbf{Z})^T - (\mathbf{e}_i^T \mathbf{Z}')^T\| \right] = \left( \frac{c \gamma_1}{\lambda} + c_1 \right) \sum_{i=1}^{m-1} \|\mathbf{e}_i^T (\mathbf{Z} - \mathbf{Z}')\| \\
& = \left( \frac{c \gamma_1}{\lambda} + c_1 \right) \sum_{i=1}^{m-1} \|(\mathbf{e}_i^T \mathbf{P}^K (\mathbf{X} - \mathbf{X}'))\| \\
& = \left( \frac{c \gamma_1}{\lambda} + c_1 \right) \sum_{i=1}^{m-1} \|(\mathbf{e}_i^T \mathbf{P}^K \tilde{\mathbf{D}}^{-1} \tilde{\mathbf{D}}(\mathbf{X} - \mathbf{X}'))\| \\
& \overset{(a)}{=} \left( \frac{c \gamma_1}{\lambda} + c_1 \right) \sum_{i=1}^{m-1} \|(\mathbf{e}_i^T \mathbf{P}^K \tilde{\mathbf{D}}^{-1} \tilde{\mathbf{D}} \mathbf{e}_m \mathbf{e}_m^T \mathbf{X})\| \\
& \overset{(b)}{\leq} \left( \frac{c \gamma_1}{\lambda} + c_1 \right) \sum_{i=1}^{m-1} \|\mathbf{e}_i^T \mathbf{P}^K \tilde{\mathbf{D}}^{-1} \tilde{\mathbf{D}} \mathbf{e}_m\| \|\mathbf{e}_m^T \mathbf{X}\| \\
& \overset{(c)}{\leq} \left( \frac{c \gamma_1}{\lambda} + c_1 \right) \sum_{i=1}^{m-1} \|\mathbf{e}_i^T \mathbf{P}^K \tilde{\mathbf{D}}^{-1} \tilde{\mathbf{D}} \mathbf{e}_m\| \\
& = \left( \frac{c \gamma_1}{\lambda} + c_1 \right) \sum_{i=1}^{m-1} \|\mathbf{e}_i^T \mathbf{P}^K \tilde{\mathbf{D}}^{-1} \mathbf{e}_m \tilde{\mathbf{D}}_{mm}\| \\
& \overset{(d)}{=} \left( \frac{c \gamma_1}{\lambda} + c_1 \right) \sum_{i=1}^{m-1} \mathbf{e}_i^T \mathbf{P}^K \tilde{\mathbf{D}}^{-1} \mathbf{e}_m \tilde{\mathbf{D}}_{mm} \\
& \overset{(e)}{\leq} \left( \frac{c \gamma_1}{\lambda} + c_1 \right) \mathbf{1}^T \mathbf{P}^K \tilde{\mathbf{D}}^{-1} \mathbf{e}_m \tilde{\mathbf{D}}_{mm} \\
& \overset{(f)}{=} \left( \frac{c \gamma_1}{\lambda} + c_1 \right) \mathbf{1}^T \tilde{\mathbf{D}}^{-1} \mathbf{p} \tilde{\mathbf{D}}_{mm} \\
& \overset{(g)}{\leq} \left( \frac{c \gamma_1}{\lambda} + c_1 \right) \tilde{\mathbf{D}}_{mm}.
\end{aligned}
\tag{21}
$$

Inequality (a) follows from the fact that $\mathbf{X}'$ is identical to $\mathbf{X}$ except for the last row and column, which are set to all-zeros. Thus, $\mathbf{X} - \mathbf{X}'$ is a matrix with rows equal to zero-vectors, except for the $m^{th}$ which equals the $m^{th}$ row of $\mathbf{X}$. Inequality (b) follows from the Cauchy-Schwartz inequality. Inequality (c) is a result of (3) in Assumption 4.2, while (d) is a consequence of the fact that $\mathbf{e}_i^T \mathbf{P}^K \tilde{\mathbf{D}}^{-1} \mathbf{e}_m$ is the value in the $i^{th}$ row and $m^{th}$ column of the matrix $\mathbf{P}^K \tilde{\mathbf{D}}^{-1}$. Also, it is obvious that this matrix is entry-wise nonnegative. Inequality (e) is due to the fact that $\mathbf{P}^K \tilde{\mathbf{D}}^{-1}$ is entry-wise nonnegative. In (f), $\mathbf{p}$ stands for a probability vector and (f) holds since

$$\mathbf{P}^K \tilde{\mathbf{D}}^{-1} = \left(\tilde{\mathbf{D}}^{-1}\tilde{\mathbf{A}}\right)^K \tilde{\mathbf{D}}^{-1} = \tilde{\mathbf{D}}^{-1}(\tilde{\mathbf{A}}\tilde{\mathbf{D}}^{-1})^K, \tag{22}$$

and $\tilde{\mathbf{A}}\tilde{\mathbf{D}}^{-1}$ is a left stochastic matrix. Inequality (g) is a consequence of the observation that the maximum entry in $\tilde{\mathbf{D}}^{-1}$ is at most 1 and that the latter is a diagonal matrix. Hence, $\mathbf{1}^T\tilde{\mathbf{D}}^{-1}\mathbf{p} \le \mathbf{1}^T\mathbf{p}$. Also, $\mathbf{1}^T\mathbf{p} = 1$ by the definition of the probability vector.

Combining the bounds, we obtain

$$\|\Delta\| \le c + c + \left(\frac{c\gamma_1}{\lambda} + c_1\right)\tilde{\mathbf{D}}_{mm} = \frac{2c\lambda + (c\gamma_1 + \lambda c_1)\tilde{\mathbf{D}}_{mm}}{\lambda}. \tag{23}$$

Including the bound on $\|\mathbf{H}_{\mathbf{w}^\star}^{-1}\|$ and equation 12, we then obtain

$$\|G(\mathbf{w}^-)\| \le \gamma_2(m-1)\|\mathbf{H}_{\mathbf{w}^\star}^{-1}\Delta\|^2 \le \gamma_2(m-1)\left(\frac{\frac{2c\lambda + (c\gamma_1 + \lambda c_1)\tilde{\mathbf{D}}_{mm}}{\lambda}}{\lambda(m-1)}\right)^2$$

$$= \frac{\gamma_2(2c\lambda + (c\gamma_1 + \lambda c_1)\tilde{\mathbf{D}}_{mm})^2}{\lambda^4(m-1)}. \tag{24}$$

This completes the proof. $\qquad\square$

## A.8 Proof of Theorem 4.4

**Theorem.** *For the edge unlearning case, we have $\mathcal{D} = (\mathbf{X}, \mathbf{Y}_{T_r}, \mathbf{P})$ and $\mathcal{D}' = (\mathbf{X}, \mathbf{Y}_{T_r}, \mathbf{P}')$. If $\mathbf{P} = \tilde{\mathbf{D}}^{-1}\tilde{\mathbf{A}}$ and $\mathbf{Z} = \mathbf{P}^K\mathbf{X}$, then we have*

$$\|\nabla L(\mathbf{w}^-, \mathcal{D}')\| = \|(\mathbf{H}_{\mathbf{w}_\eta} - \mathbf{H}_{\mathbf{w}^\star})\mathbf{H}_{\mathbf{w}^\star}^{-1}\Delta\| \le \frac{16\gamma_2 K^2 (c\gamma_1 + c_1\lambda)^2}{\lambda^4 m}. \tag{25}$$

Similar to what holds for the node feature unlearning case, Theorem 4.4 still holds when neither of the two end nodes of the removed edge belongs to $T_r$. Since $P'$ is a right stochastic matrix, Lemma A.1 still applies. Thus, we only need to describe how to bound $\|\Delta\|$. Following an approach similar to the previously described one, we have $\|\Delta\| \le \left(\frac{c\gamma_1}{\lambda} + c_1\right)\sum_{i=1}^{m}\sum_{j=1}^{n}\|\mathbf{e}_i^T(\mathbf{P}^K - \mathbf{P}'^K)\mathbf{e}_j\|$. We also need the following technical lemmas.

**Lemma A.2.** *For both edge and node unlearning, we have $|\mathbf{e}_i^T[\mathbf{P}^K - (\mathbf{P}')^K]\mathbf{e}_j| \le \sum_{k=1}^{K}\mathbf{e}_i^T(\mathbf{P}')^{k-1}|\mathbf{P} - \mathbf{P}'|\mathbf{P}^{K-k}\mathbf{e}_j, \forall i,j \in [n], K \ge 1$.*

**Lemma A.3.** *For edge unlearning, we have $\mathbf{1}^T\mathbf{P}'^{k-1}|\mathbf{P} - \mathbf{P}'|\mathbf{P}^{K-k}\mathbf{1} \le 4, \forall k \in [K]$.*

Combining the two lemmas and after some algebraic manipulation, we arrive at the desired result. It is not hard to see that $|\mathbf{P} - \mathbf{P}'|$ has only two nonzero rows, which correspond to the unlearned edge. One can again construct a left stochastic matrix $\tilde{\mathbf{A}}'\tilde{\mathbf{D}}'^{-1}$ and a right stochastic matrix $\mathbf{P}$ which lead to the result of Lemma A.3.

*Proof.* The theorem can be proved as follows. From previous proof we have

$$\|G(\mathbf{w}^-)\| \le \|\mathbf{H}_{\mathbf{w}_\eta} - \mathbf{H}_{\mathbf{w}^\star}\|\|\mathbf{H}_{\mathbf{w}^\star}^{-1}\|\|\Delta\| \le \gamma_2\frac{\|\Delta\|^2}{\lambda^2 m}. \tag{26}$$

Since the first term $\|\mathbf{H}_{\mathbf{w}_\eta} - \mathbf{H}_{\mathbf{w}^\star}\|$ only involved the updated dataset, the upper bound for this term proved for node feature unlearning still holds. The term $\|\mathbf{H}_{\mathbf{w}^\star}^{-1}\|$ can again be bounded using the fact

that $L(\cdot, \mathcal{D}')$ is $\lambda m$-strongly convex. The main difference between node feature and edge unlearning lies in the bound for $\Delta$. By definition,

$$
\begin{aligned}
\Delta =& \nabla L(\mathbf{w}^\star, \mathcal{D}) - \nabla L(\mathbf{w}^\star, \mathcal{D}') \\
=& \sum_{i=1}^m \left[ \nabla \ell(\mathbf{e}_i^T \mathbf{Z} \mathbf{w}^\star, \mathbf{e}_i^T \mathbf{Y}_{T_r}) - \nabla \ell(\mathbf{e}_i^T \mathbf{Z}' \mathbf{w}^\star, \mathbf{e}_i^T \mathbf{Y}_{T_r}) \right], \text{ and}
\end{aligned}
$$

$$
\begin{aligned}
\|\Delta\| &\leq \left( \frac{c\gamma_1}{\lambda} + c_1 \right) \sum_{i=1}^m \|(\mathbf{Z} - \mathbf{Z}')^T \mathbf{e}_i\| \\
&= \left( \frac{c\gamma_1}{\lambda} + c_1 \right) \sum_{i=1}^m \|(\mathbf{P}^K \mathbf{X} - \mathbf{P}'^K \mathbf{X})^T \mathbf{e}_i\| \\
&= \left( \frac{c\gamma_1}{\lambda} + c_1 \right) \sum_{i=1}^m \|\mathbf{e}_i^T (\mathbf{P}^K - \mathbf{P}'^K) \mathbf{X}\| \\
&= \left( \frac{c\gamma_1}{\lambda} + c_1 \right) \sum_{i=1}^m \|\mathbf{e}_i^T (\mathbf{P}^K - \mathbf{P}'^K) \sum_{j=1}^n \mathbf{e}_j \mathbf{e}_j^T \mathbf{X}\| \\
&\leq \left( \frac{c\gamma_1}{\lambda} + c_1 \right) \sum_{i=1}^m \sum_{j=1}^n \|\mathbf{e}_i^T (\mathbf{P}^K - \mathbf{P}'^K) \mathbf{e}_j \mathbf{e}_j^T \mathbf{X}\| \\
&\leq \left( \frac{c\gamma_1}{\lambda} + c_1 \right) \sum_{i=1}^m \sum_{j=1}^n \|\mathbf{e}_i^T (\mathbf{P}^K - \mathbf{P}'^K) \mathbf{e}_j\| \|\mathbf{e}_j^T \mathbf{X}\| \\
&\leq \left( \frac{c\gamma_1}{\lambda} + c_1 \right) \sum_{i=1}^m \sum_{j=1}^n \|\mathbf{e}_i^T (\mathbf{P}^K - \mathbf{P}'^K) \mathbf{e}_j\| \quad (27)
\end{aligned}
$$

By Lemma A.2 we have

$$
\begin{aligned}
&\left( \frac{c\gamma_1}{\lambda} + c_1 \right) \sum_{i=1}^m \sum_{j=1}^n \|\mathbf{e}_i^T (\mathbf{P}^K - \mathbf{P}'^K) \mathbf{e}_j\| \\
&\leq \left( \frac{c\gamma_1}{\lambda} + c_1 \right) \sum_{i=1}^m \sum_{j=1}^n \sum_{k=1}^K \mathbf{e}_i^T \mathbf{P}'^{k-1} |\mathbf{P} - \mathbf{P}'| \mathbf{P}^{K-k} \mathbf{e}_j \\
&\leq \left( \frac{c\gamma_1}{\lambda} + c_1 \right) \sum_{k=1}^K \mathbf{1}^T \mathbf{P}'^{k-1} |\mathbf{P} - \mathbf{P}'| \mathbf{P}^{K-k} \mathbf{1}. \quad (28)
\end{aligned}
$$

Using Lemma A.3 we arrive at $\|\Delta\| \leq \left( \frac{c\gamma_1}{\lambda} + c_1 \right) 4K$. Plugging this expression into equation 26 completes the proof. $\qquad \square$

## A.9 Proof of Theorem 4.5

**Theorem.** *Under the node unlearning scenario, we have $\mathcal{D} = (\mathbf{X}, \mathbf{Y}_{T_r}, \mathbf{P})$ and $\mathcal{D} = (\mathbf{X}', \mathbf{Y}'_{T_r}, \mathbf{P}')$. Suppose also that Assumption 4.2 holds. For $\mathbf{Z} = \mathbf{P}^K \mathbf{X}$ and $\mathbf{P} = \tilde{\mathbf{D}}^{-1} \tilde{\mathbf{A}}$, we have*

$$
\|\nabla L(\mathbf{w}^-, \mathcal{D}')\| = \|(\mathbf{H}_{\mathbf{w}_\eta} - \mathbf{H}_{\mathbf{w}^\star}) \mathbf{H}_{\mathbf{w}^\star}^{-1} \Delta\| \leq \frac{\gamma_2 \left( 2c\lambda + K (c\gamma_1 + c_1 \lambda) \left( 2\tilde{\mathbf{D}}_{mm} - 1 \right) \right)^2}{\lambda^4 (m-1)}. \quad (29)
$$

Again, the main challenge is to bound $\Delta$. First we observe that $(\mathbf{P}')^K \mathbf{X}' = (\mathbf{P}')^K \mathbf{X}$. This holds because node $m$ is removed from the graph in $\mathcal{D}'$, and thus its corresponding node features do not affect $\mathbf{Z}'$. Similarly to the proof Lemma A.2, we first derive the bound $\sum_{k=1}^K \mathbf{1}^T (\mathbf{P}')^{k-1} |\mathbf{P} - \mathbf{P}'| \mathbf{P}^{K-k} \mathbf{1}$. For each term, $\mathbf{1}^T (\mathbf{P}')^{k-1} |\mathbf{P} - \mathbf{P}'| \mathbf{P}^{K-k} \mathbf{1} = \sum_{l=1}^n \mathbf{1}^T (\mathbf{P}')^{k-1} \mathbf{e}_l \mathbf{e}_l^T |\mathbf{P} - \mathbf{P}'| \mathbf{P}^{K-k} \mathbf{1}$. To proceed, we need the following two lemmas.

**Lemma A.4.** *For node unlearning and* $\forall k \in [K]$ *and* $\forall l \in [n]$, $\mathbf{1}^T (\mathbf{P}')^{k-1} (\tilde{\mathbf{D}}')^{-1} \mathbf{e}_l \leq 1$.

**Lemma A.5.** *For node unlearning and* $\forall k \in [K]$, $\sum_{l=1}^{n} \mathbf{e}_l^T \tilde{\mathbf{D}}' |\mathbf{P} - \mathbf{P}'| \mathbf{P}^{K-k} \mathbf{1} \leq 2 \tilde{\mathbf{D}}_{mm} - 1$.

These two lemmas give rise to the term $K(2\tilde{\mathbf{D}}_{mm} - 1)$ in the bound of Theorem 4.5 and the rest of the analysis is similar to that of the previous cases. Lemma A.5 is rather technical, and relies on the following proposition that exploits the structure of $|\mathbf{P} - \mathbf{P}'|$.

**Proposition A.6.** *For node unlearning and* $\forall i, j \neq m$, $\mathbf{e}_i^T |\mathbf{P} - \mathbf{P}'| \mathbf{e}_j = \mathbf{e}_i^T (\mathbf{P}' - \mathbf{P}) \mathbf{e}_j$. *For* $i = m$ *or* $j = m$, $\mathbf{e}_i^T |\mathbf{P} - \mathbf{P}'| \mathbf{e}_j = \mathbf{e}_i^T \mathbf{P} \mathbf{e}_j$.

*Proof.* The proof is similar to the proof of Theorem 4.3, although several parts need modifications. First, the result of Lemma A.1 needs to be replaced by the following claim.

**Lemma A.7.** *Assume that* $\|\mathbf{e}_i^T \mathbf{S}\| \leq 1$, $\forall i \neq m$ *and that* $\mathbf{e}_m^T \mathbf{S} = \mathbf{0}^T$. *Then* $\forall i \in [n]$, $K \geq 0$, *we have* $\|\mathbf{e}_i^T (\mathbf{P}')^K \mathbf{S}\| \leq 1$, *where* $\mathbf{P} = \tilde{\mathbf{D}}^{-1} \tilde{\mathbf{A}}$ *and* $\mathbf{P}' = (\tilde{\mathbf{D}}')^{-1} \tilde{\mathbf{A}}'$.

$\square$

Next we have to modify the proof regarding the bound of $\|\Delta\|$. Following a proof similar to that of Theorem 4.3, we have

$$\|\Delta\| \leq 2c + \left( \frac{c\gamma_1}{\lambda} + c_1 \right) \sum_{i=1}^{m-1} \|(\mathbf{Z} - \mathbf{Z}')^T \mathbf{e}_i\|. \tag{30}$$

Plugging in the expressions for $\mathbf{Z}$ and $\mathbf{Z}'$ leads to

$$
\begin{aligned}
\sum_{i=1}^{m-1} \|(\mathbf{Z} - \mathbf{Z}')^T \mathbf{e}_i\| &= \sum_{i=1}^{m-1} \|(\mathbf{P}^K \mathbf{X} - (\mathbf{P}')^K \mathbf{X}')^T \mathbf{e}_i\| \\
&\overset{(a)}{=} \sum_{i=1}^{m-1} \|(\mathbf{P}^K \mathbf{X} - (\mathbf{P}')^K \mathbf{X})^T \mathbf{e}_i\| = \sum_{i=1}^{m-1} \|\left( \left[ \mathbf{P}^K - (\mathbf{P}')^K \right] \mathbf{X} \right)^T \mathbf{e}_i\| \\
&\overset{(b)}{=} \sum_{i=1}^{m-1} \|\mathbf{e}_i^T \left[ \mathbf{P}^K - (\mathbf{P}')^K \right] \sum_{j=1}^{n} \mathbf{e}_j \mathbf{e}_j^T \mathbf{X}\| \\
&\overset{(c)}{\leq} \sum_{i=1}^{m-1} \sum_{j=1}^{n} \|\mathbf{e}_i^T \left[ \mathbf{P}^K - (\mathbf{P}')^K \right] \mathbf{e}_j \mathbf{e}_j^T \mathbf{X}\| \\
&\overset{(d)}{\leq} \sum_{i=1}^{m-1} \sum_{j=1}^{n} \|\mathbf{e}_i^T \left[ \mathbf{P}^K - (\mathbf{P}')^K \right] \mathbf{e}_j\| \|\mathbf{e}_j^T \mathbf{X}\| \\
&\overset{(e)}{\leq} \sum_{i=1}^{m-1} \sum_{j=1}^{n} \|\mathbf{e}_i^T \left[ \mathbf{P}^K - (\mathbf{P}')^K \right] \mathbf{e}_j\|. \tag{31}
\end{aligned}
$$

The equality (a) is due to the fact that $(\mathbf{P}')^K \mathbf{X}' = (\mathbf{P}')^K \mathbf{X}$, as the $m^{th}$ row and column of $(\mathbf{P}')^K$ are all-zeros. Thus, changing the last row of $\mathbf{X}'$ makes no difference of $(\mathbf{P}')^K \mathbf{X}'$. Equation (b) is a consequence of the fact that $\mathbf{I} = \sum_{j=1}^{n} \mathbf{e}_j \mathbf{e}_j^T$. Inequality (c) follows from Minkowski's inequality, while (d) follows from the Cauchy-Schwartz inequality. Inequality (e) holds based on (3) in Assumption 4.2.

By Lemma A.2, we can proceed with our analysis as follows:

$$\sum_{i=1}^{m-1} \|(\mathbf{Z} - \mathbf{Z}')^T \mathbf{e}_i\|$$

$$\leq \sum_{i=1}^{m-1} \sum_{j=1}^{n} \|\mathbf{e}_i^T \left[ \mathbf{P}^K - (\mathbf{P}')^K \right] \mathbf{e}_j\|$$

$$\overset{(a)}{\leq} \sum_{i=1}^{m-1} \sum_{j=1}^{n} \sum_{k=1}^{K} \mathbf{e}_i^T (\mathbf{P}')^{k-1} |\mathbf{P} - \mathbf{P}'| \mathbf{P}^{K-k} \mathbf{e}_j$$

$$\leq \sum_{k=1}^{K} \mathbf{1}^T (\mathbf{P}')^{k-1} |\mathbf{P} - \mathbf{P}'| \mathbf{P}^{K-k} \mathbf{1}, \tag{32}$$

where (a) is due to Lemma A.2 and the fact that $\mathbf{e}_i^T \mathbf{P} \mathbf{e}_j$ is a scalar, equal to the $i^{th}$ row $j^{th}$ column of the matrix $\mathbf{P}$.

Next, we bound each term $\mathbf{1}^T (\mathbf{P}')^{k-1} |\mathbf{P} - \mathbf{P}'| \mathbf{P}^{K-k} \mathbf{1}$ separately. For $k \in [K]$, we have

$$\mathbf{1}^T (\mathbf{P}')^{k-1} |\mathbf{P} - \mathbf{P}'| \mathbf{P}^{K-k} \mathbf{1}$$

$$= \mathbf{1}^T (\mathbf{P}')^{k-1} (\tilde{\mathbf{D}}')^{-1} \tilde{\mathbf{D}}' |\mathbf{P} - \mathbf{P}'| \mathbf{P}^{K-k} \mathbf{1}$$

$$= \mathbf{1}^T (\mathbf{P}')^{k-1} (\tilde{\mathbf{D}}')^{-1} \sum_{l=1}^{n} \mathbf{e}_l \mathbf{e}_l^T \tilde{\mathbf{D}}' |\mathbf{P} - \mathbf{P}'| \mathbf{P}^{K-k} \mathbf{1}$$

$$= \sum_{l=1}^{n} \left( \mathbf{1}^T (\mathbf{P}')^{k-1} (\tilde{\mathbf{D}}')^{-1} \mathbf{e}_l \right) \left( \mathbf{e}_l^T \tilde{\mathbf{D}}' |\mathbf{P} - \mathbf{P}'| \mathbf{P}^{K-k} \mathbf{1} \right). \tag{33}$$

Note that for each index $l$, the corresponding term in the sum is just a product of two scalars. Let first analyze $\mathbf{1}^T (\mathbf{P}')^{k-1} (\hat{\mathbf{D}}')^{-1} \mathbf{e}_l$. This term can be bounded as

$$\mathbf{1}^T (\mathbf{P}')^{k-1} |\mathbf{P} - \mathbf{P}'| \mathbf{P}^{K-k} \mathbf{1}$$

$$= \sum_{j=1}^{n} \left( \mathbf{1}^T (\mathbf{P}')^{k-1} (\tilde{\mathbf{D}}')^{-1} \mathbf{e}_j \right) \left( \mathbf{e}_j^T \tilde{\mathbf{D}}' |\mathbf{P} - \mathbf{P}'| \mathbf{P}^{K-k} \mathbf{1} \right)$$

$$\overset{(a)}{\leq} \sum_{j=1}^{n} \mathbf{e}_j^T \tilde{\mathbf{D}}' |\mathbf{P} - \mathbf{P}'| \mathbf{P}^{K-k} \mathbf{1}. \tag{34}$$

where (a) follows from Lemma A.4.

We now turn our attention to the term $\mathbf{e}_l^T \tilde{\mathbf{D}}' |\mathbf{P} - \mathbf{P}'| \mathbf{P}^{K-k} \mathbf{1}$, which can be bounded according to Lemma A.5 as follows

$$\mathbf{1}^T (\mathbf{P}')^{k-1} |\mathbf{P} - \mathbf{P}'| \mathbf{P}^{K-k} \mathbf{1}$$

$$\leq \sum_{l=1}^{n} \mathbf{e}_l^T \tilde{\mathbf{D}}' |\mathbf{P} - \mathbf{P}'| \mathbf{P}^{K-k} \mathbf{1}$$

$$\leq 2 \tilde{\mathbf{D}}_{mm} - 1. \tag{35}$$

Using these two bounds in equation 32 gives

$$
\begin{aligned}
\sum_{i=1}^{m-1} & \|(\mathbf{Z} - \mathbf{Z}')^T \mathbf{e}_i\| \\
&\leq \sum_{i=1}^{m-1} \sum_{j=1}^{n} \|\mathbf{e}_i^T \left[ \mathbf{P}^K - (\mathbf{P}')^K \right] \mathbf{e}_j\| \\
&\leq \sum_{k=1}^{K} \mathbf{1}^T (\mathbf{P}')^{k-1} |\mathbf{P} - \mathbf{P}'| \, \mathbf{P}^{K-k} \mathbf{1} \\
&\leq \sum_{k=1}^{K} (2\tilde{\mathbf{D}}_{mm} + 1) = K(2\tilde{\mathbf{D}}_{mm} - 1).
\end{aligned}
\tag{36}
$$

Using this bound in the expression for $\|\Delta\|$ we obtain

$$
\begin{aligned}
\|\Delta\| &\leq 2c + \left( \frac{c\gamma_1}{\lambda} + c_1 \right) \sum_{i=1}^{m-1} \|(\mathbf{Z} - \mathbf{Z}')^T \mathbf{e}_i\| \\
&\leq 2c + \left( \frac{c\gamma_1}{\lambda} + c_1 \right) K \left( 2\tilde{\mathbf{D}}_{mm} - 1 \right) \\
\Rightarrow \|G(\mathbf{w}^-)\| &\leq \|\mathbf{H}_{\mathbf{w}_\eta} - \mathbf{H}_{\mathbf{w}^\star}\| \|\mathbf{H}_{\mathbf{w}^\star}^{-1} \Delta\| \\
&\leq \gamma_2 (m-1) \|\mathbf{H}_{\mathbf{w}^\star}^{-1} \Delta\|^2 \\
&\leq \gamma_2 (m-1) \left( \frac{2c + \left( \frac{c\gamma_1}{\lambda} + c_1 \right) K \left( 2\tilde{\mathbf{D}}_{mm} - 1 \right)}{\lambda(m-1)} \right)^2 \\
&= \frac{\gamma_2 \left( 2c\lambda + K \left( c\gamma_1 + c_1\lambda \right) \left( 2\tilde{\mathbf{D}}_{mm} - 1 \right) \right)^2}{\lambda^4 (m-1)}.
\end{aligned}
\tag{37}
$$

This completes the proof.

## A.10 PROOF OF THEOREM 4.6

**Theorem.** *In the node feature unlearning scenario, we are given $\mathcal{D} = (\mathbf{X}, \mathbf{Y}_{T_r}, \mathbf{A})$ and $\mathcal{D} = (\mathbf{X}', \mathbf{Y}'_{T_r}, \mathbf{A})$. Suppose that Assumption 4.2 holds. For $\mathbf{Z} = \frac{1}{K+1} \left[ \mathbf{X}, \mathbf{PX}, \mathbf{P}^2\mathbf{X}, \cdots, \mathbf{P}^K\mathbf{X} \right]$ and $\mathbf{P} = \tilde{\mathbf{D}}^{-1}\tilde{\mathbf{A}}$, we have*

$$
\|\nabla L(\mathbf{w}^-, \mathcal{D}')\| = \|(\mathbf{H}_{\mathbf{w}_\eta} - \mathbf{H}_{\mathbf{w}^\star}) \mathbf{H}_{\mathbf{w}^\star}^{-1} \Delta\| \leq \frac{\gamma_2 (2c\lambda + (c\gamma_1 + \lambda c_1) \tilde{\mathbf{D}}_{mm})^2}{\lambda^4 (m-1)}.
\tag{38}
$$

*Proof.* The proof is almost identical to the proof of Theorem 4.3. We only need to bound the norms of the terms in $\mathbf{Z}$. We start by modifying Lemma A.1 for the GPR case.

**Lemma A.8.** *Assume that $\|\mathbf{e}_i^T \mathbf{S}\| \leq 1$, $\forall i \in [n]$. Then $\forall i \in [n]$, $K \geq 0$, we have $\|\frac{1}{\sqrt{K+1}} \mathbf{e}_i^T \left[ \mathbf{S}, \mathbf{PS}, \mathbf{P}^2\mathbf{S}, \cdots, \mathbf{P}^K\mathbf{S} \right]\| \leq 1$, where $\mathbf{P} = \tilde{\mathbf{D}}^{-1}\tilde{\mathbf{A}}$.*

Another part of the proof that needs to be changed is to establish a bound on

$$
\sum_{i=1}^{m-1} \left[ \nabla\ell(\mathbf{e}_i^T \mathbf{Z}\mathbf{w}^\star, \mathbf{e}_i^T \mathbf{Y}_{T_r}) - \nabla\ell(\mathbf{e}_i^T \mathbf{Z}'\mathbf{w}^\star, \mathbf{e}_i^T \mathbf{Y}_{T_r}) \right].
\tag{39}
$$

Following a proof similar to that of Theorem 4.3, we have

$$\| \sum_{i=1}^{m-1} \left[ \nabla\ell(\mathbf{e}_i^T\mathbf{Z}\mathbf{w}^\star, \mathbf{e}_i^T\mathbf{Y}_{T_r}) - \nabla\ell(\mathbf{e}_i^T\mathbf{Z}'\mathbf{w}^\star, \mathbf{e}_i^T\mathbf{Y}_{T_r}) \right] \|$$

$$\leq \sum_{i=1}^{m-1} \left[ \left( \frac{c\gamma_1}{\lambda} + c_1 \right) \|(\mathbf{e}_i^T\mathbf{Z})^T - (\mathbf{e}_i^T\mathbf{Z}')^T\| \right]$$

$$= \left( \frac{c\gamma_1}{\lambda} + c_1 \right) \sum_{i=1}^{m-1} \|(\mathbf{Z} - \mathbf{Z}')^T\mathbf{e}_i\|$$

$$= \left( \frac{c\gamma_1}{\lambda} + c_1 \right) \sum_{i=1}^{m-1} \|(\frac{1}{K+1} \left[ \mathbf{X} - \mathbf{X}', \mathbf{P}(\mathbf{X} - \mathbf{X}'), \cdots, \mathbf{P}^K(\mathbf{X} - \mathbf{X}') \right])^T\mathbf{e}_i\|$$

$$= \left( \frac{c\gamma_1}{\lambda} + c_1 \right) \sum_{i=1}^{m-1} \|(\frac{1}{K+1} \left[ \mathbf{e}_m\mathbf{e}_m^T\mathbf{X}, \mathbf{P}\mathbf{e}_m\mathbf{e}_m^T\mathbf{X}, \cdots, \mathbf{P}^K\mathbf{e}_m\mathbf{e}_m^T\mathbf{X} \right])^T\mathbf{e}_i\|$$

$$= \left( \frac{c\gamma_1}{\lambda} + c_1 \right) \sum_{i=1}^{m-1} \|\frac{1}{K+1} \left[ \mathbf{e}_i^T\mathbf{e}_m\mathbf{e}_m^T\mathbf{X}, \mathbf{e}_i^T\mathbf{P}\mathbf{e}_m\mathbf{e}_m^T\mathbf{X}, \cdots, \mathbf{e}_i^T\mathbf{P}^K\mathbf{e}_m\mathbf{e}_m^T\mathbf{X} \right]^T \|$$

$$\leq \left( \frac{c\gamma_1}{\lambda} + c_1 \right) \sum_{i=1}^{m-1} \|\frac{1}{K+1} \left[ \mathbf{e}_i^T\mathbf{e}_m, \mathbf{e}_i^T\mathbf{P}\mathbf{e}_m, \cdots, \mathbf{e}_i^T\mathbf{P}^K\mathbf{e}_m \right]^T \| \|(\mathbf{e}_m^T\mathbf{X})^T\|$$

$$\leq \left( \frac{c\gamma_1}{\lambda} + c_1 \right) \sum_{i=1}^{m-1} \|\frac{1}{K+1} \left[ \mathbf{e}_i^T\mathbf{e}_m, \mathbf{e}_i^T\mathbf{P}\mathbf{e}_m, \cdots, \mathbf{e}_i^T\mathbf{P}^K\mathbf{e}_m \right]^T \|$$

$$\overset{(a)}{\leq} \left( \frac{c\gamma_1}{\lambda} + c_1 \right) \sum_{i=1}^{m-1} \frac{1}{K+1} \sum_{k=1}^{K} \mathbf{e}_i^T\mathbf{P}^k\mathbf{e}_m$$

$$\leq \frac{\frac{c\gamma_1}{\lambda} + c_1}{K+1} \sum_{k=1}^{K} \mathbf{1}^T\mathbf{P}^k\mathbf{e}_m = \frac{\frac{c\gamma_1}{\lambda} + c_1}{K+1} \sum_{k=1}^{K} \mathbf{1}^T\mathbf{P}^k\tilde{\mathbf{D}}^{-1}\tilde{\mathbf{D}}\mathbf{e}_m = \frac{\frac{c\gamma_1}{\lambda} + c_1}{K+1} \sum_{k=1}^{K} \mathbf{1}^T\mathbf{P}^k\tilde{\mathbf{D}}^{-1}\mathbf{e}_m\tilde{\mathbf{D}}_{mm}$$

$$\overset{(b)}{=} \frac{\frac{c\gamma_1}{\lambda} + c_1}{K+1} \sum_{k=1}^{K} \mathbf{1}^T\tilde{\mathbf{D}}^{-1}\mathbf{p}^{(k)}\tilde{\mathbf{D}}_{mm}$$

$$\leq \frac{\frac{c\gamma_1}{\lambda} + c_1}{K+1} \sum_{k=1}^{K} \mathbf{1}^T\mathbf{p}^{(k)}\tilde{\mathbf{D}}_{mm} = \frac{\frac{c\gamma_1}{\lambda} + c_1}{K+1} \times K\tilde{\mathbf{D}}_{mm}$$

$$\leq (\frac{c\gamma_1}{\lambda} + c_1)\tilde{\mathbf{D}}_{mm}, \tag{40}$$

where (a) is due to the fact that the $\ell_1$ norm is an upper bound for the $\ell_2$ norm. Also note that $\mathbf{e}_i^T\mathbf{e}_m = 0, \forall i \neq m$. In (b), $\forall k \in [K]$, $\mathbf{p}^{(k)}$ are probability vectors. This completes the proof. $\qquad\square$

*Remark.* Note that the GPR extension for the edge and node unlearning cases can be derived through a similar analysis. One can also see that the key step is inequality (a), which still holds for the edge and node unlearning cases. The results are similar to Theorem 4.4 and Theorem 4.5, except that the definition of $\mathbf{Z}$ is replaced by one corresponding to the GPR case, as in Theorem 4.6.

### A.11 PROOF OF LEMMA A.1

**Lemma.** *Assume that $\|\mathbf{e}_i^T\mathbf{S}\| \leq 1$, $\forall i \in [n]$. Then $\forall i \in [n]$, $K \geq 0$, we have $\|\mathbf{e}_i^T\mathbf{P}^K\mathbf{S}\| \leq 1$, where $\mathbf{P} = \tilde{\mathbf{D}}^{-1}\tilde{\mathbf{A}}$.*

*Proof.* We prove this lemma by induction. Let $\mathbf{Z}^{(k)} = \mathbf{P}^k\mathbf{S}$. For the base case $k = 0$ it is true by assumption that $\|\mathbf{e}_i^T\mathbf{S}\| \leq 1 \ \forall i \in [n]$. Assume next that the claim is true for the case $k = K - 1$.

Then we have

$$\|\mathbf{e}_i^T \mathbf{P}^K \mathbf{S}\| = \|\mathbf{e}_i^T \mathbf{P} \mathbf{Z}^{(K-1)}\| = \|\frac{1}{\tilde{\mathbf{D}}_{ii}} \sum_{j:\tilde{\mathbf{A}}_{ij}=1} \mathbf{e}_j^T \mathbf{Z}^{(K-1)}\| \leq \frac{1}{\tilde{\mathbf{D}}_{ii}} \sum_{j:\tilde{\mathbf{A}}_{ij}=1} \|\mathbf{e}_j^T \mathbf{Z}^{(K-1)}\|$$

$$\overset{(a)}{\leq} \frac{1}{\tilde{\mathbf{D}}_{ii}} \sum_{j:\tilde{\mathbf{A}}_{ij}=1} 1 = \frac{1}{\tilde{\mathbf{D}}_{ii}} \times \tilde{\mathbf{D}}_{ii} = 1, \tag{41}$$

where (a) is based on the induction hypothesis for $k = K - 1$. □

*Remark:* Note that if we choose another propagation matrix $\mathbf{P}$ compared to the one used in the SGC analysis, the above expression for $K = 1$ becomes

$$\|\mathbf{e}_i^T \mathbf{P} \mathbf{S}\| = \|\frac{1}{\sqrt{\tilde{\mathbf{D}}_{ii}}} \sum_{j:\tilde{\mathbf{A}}_{ij}=1} \frac{\mathbf{e}_j^T \mathbf{S}}{\sqrt{\tilde{\mathbf{D}}_{jj}}}\| \leq \frac{1}{\sqrt{\tilde{\mathbf{D}}_{ii}}} \sum_{j:\tilde{\mathbf{A}}_{ij}=1} \frac{\|\mathbf{e}_j^T \mathbf{S}\|}{\sqrt{\tilde{\mathbf{D}}_{jj}}}$$

$$\leq \frac{1}{\sqrt{\tilde{\mathbf{D}}_{ii}}} \sum_{j:\tilde{\mathbf{A}}_{ij}=1} \frac{1}{\sqrt{\tilde{\mathbf{D}}_{jj}}}. \tag{42}$$

We cannot easily simplify the sum $\sum_{j:\tilde{\mathbf{A}}_{ij}=1} \frac{1}{\sqrt{\tilde{\mathbf{D}}_{jj}}}$. One way to approach the problem is to simply use the fact that the degree of a node is at least 1 and can thusbe further upper bounded by $\tilde{\mathbf{D}}_{ii}$. This leads to the bound

$$\|\mathbf{e}_i^T \mathbf{P} \mathbf{S}\| \leq \sqrt{\mathbf{D}_{ii}}. \tag{43}$$

Obviously, this bound is worse than the one in Lemma A.1 even when $K = 1$. For general $K$, there will be an additional exponent $K/2$ for the maximal degree, which is undesirable. Nevertheless, our bound is tight since for the worst case of a star graph with a center at node $i$, so that $\mathbf{D}_{jj} = 2$ for all $j \neq i$. The same argument applies for other degree normalizations. Thus it is critical to choose $\mathbf{P} = \tilde{\mathbf{D}}^{-1} \tilde{\mathbf{A}}$ to obtained the desired bound in Lemma A.1.

## A.12 PROOF OF LEMMA A.2

**Lemma.** *For either the edge or node unlearning case, and* $\forall i, j \in [n]$, $K \geq 1$, *we have*

$$|\mathbf{e}_i^T [\mathbf{P}^K - (\mathbf{P}')^K] \mathbf{e}_j| \leq \sum_{k=1}^{K} \mathbf{e}_i^T (\mathbf{P}')^{k-1} |\mathbf{P} - \mathbf{P}'| \mathbf{P}^{K-k} \mathbf{e}_j. \tag{44}$$

*Proof.* The proof consist of two parts. We first show that

$$\mathbf{P}^K - (\mathbf{P}')^K = \sum_{k=1}^{K} (\mathbf{P}')^{k-1} (\mathbf{P} - \mathbf{P}') \mathbf{P}^{K-k}.$$

Then we proceed to analyze the absolute values of all terms in the sum.

The proof of the first part follows from a telescoping property for the sum,

$$\sum_{k=1}^{K} (\mathbf{P}')^{k-1} (\mathbf{P} - \mathbf{P}') \mathbf{P}^{K-k} = \sum_{k=1}^{K} (\mathbf{P}')^{k-1} \mathbf{P}^{K-k+1} - (\mathbf{P}')^k \mathbf{P}^{K-k}$$

$$= (\mathbf{P}')^0 \mathbf{P}^K - (\mathbf{P}')^1 \mathbf{P}^{K-1} + (\mathbf{P}')^1 \mathbf{P}^{K-1} - (\mathbf{P}')^2 \mathbf{P}^{K-2} + \cdots + (\mathbf{P}')^{K-1} \mathbf{P}^1 - (\mathbf{P}')^K \mathbf{P}^0$$

$$= \mathbf{P}^K - (\mathbf{P}')^K. \tag{45}$$

Next, note that both $\mathbf{P}'$ and $\mathbf{P}$ are nonnegative matrices, and the same is true of their $k^{th}$ powers, $k \geq 2$. Thus,

$$|\mathbf{e}_i^T [\mathbf{P}^K - (\mathbf{P}')^K] \mathbf{e}_j| = \left| \sum_{k=1}^{K} \mathbf{e}_i^T (\mathbf{P}')^{k-1} (\mathbf{P} - \mathbf{P}') \mathbf{P}^{K-k} \mathbf{e}_j \right|$$

$$\leq \sum_{k=1}^{K} \mathbf{e}_i^T (\mathbf{P}')^{k-1} |\mathbf{P} - \mathbf{P}'| \mathbf{P}^{K-k} \mathbf{e}_j. \tag{46}$$

This completes the proof. □

### A.13 PROOF OF LEMMA A.3

**Lemma.** *For the edge unlearning scenario, and $\forall k \in [K]$, we have*

$$\mathbf{1}^T \mathbf{P}'^{k-1} |\mathbf{P} - \mathbf{P}'| \mathbf{P}^{K-k} \mathbf{1} \leq 4. \tag{47}$$

*Proof.* Let us start by analyzing the matrix $|\mathbf{P} - \mathbf{P}'| = |\tilde{\mathbf{D}}^{-1}\tilde{\mathbf{A}} - \tilde{\mathbf{D}}'^{-1}\tilde{\mathbf{A}}'|$. Note that all its rows are zeros except for the $1^{st}$ and $m^{th}$ row. The first row of the matrix equals

$$\mathbf{e}_1^T |\mathbf{P} - \mathbf{P}'|$$

$$= \left[ \left( \frac{1}{d_1 - 1} - \frac{1}{d_1} \right) \tilde{\mathbf{A}}_{11}, \ldots, \left( \frac{1}{d_1 - 1} - \frac{1}{d_1} \right) \tilde{\mathbf{A}}_{1(m-1)}, \frac{1}{d_1}, \left( \frac{1}{d_1 - 1} - \frac{1}{d_1} \right) \tilde{\mathbf{A}}_{1(m+1)}, \ldots \right]$$

$$= \left[ \left( \frac{1}{d_1(d_1 - 1)} \right) \tilde{\mathbf{A}}_{11}, \ldots, \left( \frac{1}{d_1(d_1 - 1)} \right) \tilde{\mathbf{A}}_{1(m-1)}, \frac{1}{d_1}, \left( \frac{1}{d_1(d_1 - 1)} \right) \tilde{\mathbf{A}}_{1(m+1)}, \ldots \right]$$

$$= \left[ \left( \frac{1}{d_1(d_1 - 1)} \right) \tilde{\mathbf{A}}_{11}, \ldots, \left( \frac{1}{d_1(d_1 - 1)} \right) \tilde{\mathbf{A}}_{1(m-1)}, \frac{1}{d_1(d_1 - 1)}, \left( \frac{1}{d_1(d_1 - 1)} \right) \tilde{\mathbf{A}}_{1(m+1)}, \ldots \right]$$

$$+ \frac{d_1 - 2}{d_1(d_1 - 1)} \mathbf{e}_m^T = \mathbf{e}_1^T \tilde{\mathbf{D}}'^{-1} \tilde{\mathbf{D}}^{-1} \tilde{\mathbf{A}} + \frac{d_1 - 2}{d_1(d_1 - 1)} \mathbf{e}_m^T, \tag{48}$$

where the last equality holds since $\tilde{\mathbf{A}}_{1m} = 1$. Similar arguments apply for the $m^{th}$ row, for which we have

$$\mathbf{e}_m^T |\mathbf{P} - \mathbf{P}'| = \mathbf{e}_m^T \tilde{\mathbf{D}}'^{-1} \tilde{\mathbf{D}}^{-1} \tilde{\mathbf{A}} + \frac{d_m - 2}{d_m(d_m - 1)} \mathbf{e}_1^T. \tag{49}$$

For a fixed $k \in [K]$,

$$\mathbf{1}^T \mathbf{P}'^{k-1} |\mathbf{P} - \mathbf{P}'| \mathbf{P}^{K-k} \mathbf{1} = \mathbf{1}^T \mathbf{P}'^{k-1} \mathbf{e}_1 \mathbf{e}_1^T \tilde{\mathbf{D}}'^{-1} \tilde{\mathbf{D}}^{-1} \tilde{\mathbf{A}} \mathbf{P}^{K-k} \mathbf{1}$$

$$+ \mathbf{1}^T \mathbf{P}'^{k-1} \frac{d_1 - 2}{d_1(d_1 - 1)} \mathbf{e}_1 \mathbf{e}_m^T \mathbf{P}^{K-k} \mathbf{1}$$

$$+ \mathbf{1}^T \mathbf{P}'^{k-1} \mathbf{e}_m \mathbf{e}_m^T \tilde{\mathbf{D}}'^{-1} \tilde{\mathbf{D}}^{-1} \tilde{\mathbf{A}} \mathbf{P}^{K-k} \mathbf{1}$$

$$+ \mathbf{1}^T \mathbf{P}'^{k-1} \frac{d_m - 2}{d_m(d_m - 1)} \mathbf{e}_m \mathbf{e}_1^T \mathbf{P}^{K-k} \mathbf{1}. \tag{50}$$

We analyze these four terms separately. For the first term, we have

$$\mathbf{1}^T \mathbf{P}'^{k-1} \mathbf{e}_1 \mathbf{e}_1^T \tilde{\mathbf{D}}'^{-1} \tilde{\mathbf{D}}^{-1} \tilde{\mathbf{A}} \mathbf{P}^{K-k} \mathbf{1}$$

$$= \mathbf{1}^T \mathbf{P}'^{k-1} \tilde{\mathbf{D}}'^{-1} \mathbf{e}_1 \mathbf{e}_1^T \tilde{\mathbf{D}}^{-1} \tilde{\mathbf{A}} \mathbf{P}^{K-k} \mathbf{1}$$

$$= \mathbf{1}^T \mathbf{P}'^{k-1} \tilde{\mathbf{D}}'^{-1} \mathbf{e}_1 \mathbf{e}_1^T \mathbf{P}^{K-k+1} \mathbf{1} \tag{51}$$

By the same argument as used in the proof for node feature unlearning, $\mathbf{1}^T \mathbf{P}'^{k-1} \tilde{\mathbf{D}}'^{-1} \mathbf{e}_1 = \mathbf{1}^T \tilde{\mathbf{D}}'^{-1} \mathbf{p} \leq 1$, for some probability vector $\mathbf{p}$. Also, $\mathbf{e}_1^T \mathbf{P}^{K-k+1} \mathbf{1} \leq 1$, which holds due to the fact that $\mathbf{P}$ is a right-stochastic matrix. We have hence shown that the first term in equation 51 is bounded by 1. For the second term, note that $\frac{d_1 - 2}{d_1(d_1 - 1)} \leq \frac{1}{(d_1 - 1)}$. Hence,

$$\mathbf{1}^T \mathbf{P}'^{k-1} \frac{d_1 - 2}{d_1(d_1 - 1)} \mathbf{e}_1 \mathbf{e}_m^T \mathbf{P}^{K-k} \mathbf{1}$$

$$\leq \mathbf{1}^T \mathbf{P}'^{k-1} \frac{1}{d_1 - 1} \mathbf{e}_1 \mathbf{e}_m^T \mathbf{P}^{K-k} \mathbf{1}$$

$$= \mathbf{1}^T \mathbf{P}'^{k-1} \tilde{\mathbf{D}}'^{-1} \mathbf{e}_1 \mathbf{e}_m^T \mathbf{P}^{K-k} \mathbf{1} \leq 1, \tag{52}$$

where the final inequality follows the same argument as the one used for bounding the first term. For the third and fourth term, the analysis is similar to these two cases and both terms can be shown to be bounded by 1. Hence, we have

$$\mathbf{1}^T \mathbf{P}'^{k-1} |\mathbf{P} - \mathbf{P}'| \mathbf{P}^{K-k} \mathbf{1} \leq 4. \tag{53}$$

This completes the proof. □

## A.14 PROOF OF LEMMA A.4

**Lemma.** *For all $k \in [K]$ and $l \in [n]$,*

$$\mathbf{1}^T (\mathbf{P}')^{k-1} (\tilde{\mathbf{D}}')^{-1} \mathbf{e}_l \leq 1. \tag{54}$$

*Proof.* For $k = 1$, the claim is obviously true for all $l \in [n]$, as the largest entry in $\tilde{\mathbf{D}}^{-1}$ is upper bounded by 1. For $k \geq 2$ and $l \neq m$ we have

$$
\begin{aligned}
\mathbf{1}^T (\mathbf{P}')^{k-1} (\tilde{\mathbf{D}}')^{-1} \mathbf{e}_l &= \mathbf{1}^T ((\tilde{\mathbf{D}}')^{-1} \tilde{\mathbf{A}}')^{k-1} (\tilde{\mathbf{D}}')^{-1} \mathbf{e}_l \\
&= \mathbf{1}^T (\tilde{\mathbf{D}}')^{-1} (\tilde{\mathbf{A}}' (\tilde{\mathbf{D}}')^{-1})^{k-1} \mathbf{e}_l \\
&\overset{(a)}{=} \mathbf{1}^T (\tilde{\mathbf{D}}')^{-1} \mathbf{p} \leq 1.
\end{aligned} \tag{55}
$$

In (a), $\mathbf{p}$ stands for a probability vector and the result follows since $\tilde{\mathbf{A}}' (\tilde{\mathbf{D}}')^{-1}$ is a left-stochastic matrix if one ignores the node $m$. For $l = m$, it is easy to see that $\tilde{\mathbf{A}}' \tilde{\mathbf{D}}' \mathbf{e}_m = 0$ by the fact that the $m^{th}$ row and column of $\tilde{\mathbf{A}}'$ are all-zeros. This completes the proof. □

## A.15 PROOF OF LEMMA A.5

**Lemma.** *For node unlearning, and $\forall k \in [K]$, $\sum_{l=1}^{n} \mathbf{e}_l^T \tilde{\mathbf{D}}' |\mathbf{P} - \mathbf{P}'| \mathbf{P}^{K-k} \mathbf{1} \leq 2\tilde{\mathbf{D}}_{mm} - 1$.*

*Proof.* First, note that

$$\sum_{l=1}^{n} \mathbf{e}_l^T \tilde{\mathbf{D}}' |\mathbf{P} - \mathbf{P}'| = \sum_{l=1}^{n} \mathbf{e}_l^T \tilde{\mathbf{D}}' |\mathbf{P} - \mathbf{P}'| \sum_{r=1}^{n} \mathbf{e}_r \mathbf{e}_r^T = \sum_{r=1}^{n} \sum_{l=1}^{n} \mathbf{e}_l^T \tilde{\mathbf{D}}' |\mathbf{P} - \mathbf{P}'| \mathbf{e}_r \mathbf{e}_r^T. \tag{56}$$

Then for $i, j \neq m$, by Proposition A.6, we have

$$
\begin{aligned}
&\mathbf{e}_l^T \tilde{\mathbf{D}}' |\mathbf{P} - \mathbf{P}'| \mathbf{e}_r \mathbf{e}_r^T \\
&\overset{(a)}{=} \mathbf{e}_l^T \tilde{\mathbf{D}}' (\mathbf{P}' - \mathbf{P}) \mathbf{e}_r \mathbf{e}_r^T \\
&= \mathbf{e}_l^T \left( \tilde{\mathbf{A}}' - \tilde{\mathbf{D}}' \tilde{\mathbf{D}}^{-1} \tilde{\mathbf{A}} \right) \mathbf{e}_r \mathbf{e}_r^T \\
&= \left( \tilde{\mathbf{A}}'_{lr} - \frac{\tilde{\mathbf{D}}'_{ll}}{\tilde{\mathbf{D}}_{ll}} \tilde{\mathbf{A}}_{lr} \right) \mathbf{e}_r^T \\
&\overset{(b)}{=} \left( \tilde{\mathbf{A}}_{lr} - \frac{\tilde{\mathbf{D}}'_{ll}}{\tilde{\mathbf{D}}_{ll}} \tilde{\mathbf{A}}_{lr} \right) \mathbf{e}_r^T
\end{aligned} \tag{57}
$$

We used Proposition A.6 in (a) since $\mathbf{e}_l^T \tilde{\mathbf{D}}' = \tilde{\mathbf{D}}'_{ll} \mathbf{e}_l^T$. The equality (b) is due to the fact that for $i, j \neq m$, $\tilde{\mathbf{A}}'_{lr} = \tilde{\mathbf{A}}_{lr}$. Recall that $\tilde{\mathbf{A}}'$ and $\tilde{\mathbf{A}}$ only differ in the $m^{th}$ row and column.

We consider next the only two possible scenarios, (1) $l$ is a neighbor of $m$; (2) $l$ is not a neighbor of $m$. For (1), we know that $\tilde{\mathbf{D}}'_{ll} = \tilde{\mathbf{D}}_{ll} - 1 \geq 1$. This leads to

$$\tilde{\mathbf{A}}_{lr} - \frac{\tilde{\mathbf{D}}'_{ll}}{\tilde{\mathbf{D}}_{ll}} \tilde{\mathbf{A}}_{lr} = \tilde{\mathbf{A}}_{lr} \left( 1 - \frac{\tilde{\mathbf{D}}'_{ll}}{\tilde{\mathbf{D}}_{ll}} \right) = \tilde{\mathbf{A}}_{lr} \left( 1 - \frac{\tilde{\mathbf{D}}_{ll} - 1}{\tilde{\mathbf{D}}_{ll}} \right) = \frac{\tilde{\mathbf{A}}_{lr}}{\tilde{\mathbf{D}}_{ll}}. \tag{58}$$

For (2), we know that $\tilde{\mathbf{D}}'_{rr} = \tilde{\mathbf{D}}_{rr}$. Thus, $\tilde{\mathbf{A}}_{lr} - \frac{\tilde{\mathbf{D}}'_{ll}}{\tilde{\mathbf{D}}_{ll}} \tilde{\mathbf{A}}_{lr} = 0$.

Next, we consider the case $l \neq m, r = m$. Again, by Proposition A.6, we have

$$\mathbf{e}_l^T \tilde{\mathbf{D}}' |\mathbf{P} - \mathbf{P}'| \mathbf{e}_m \mathbf{e}_m^T = \mathbf{e}_l^T \tilde{\mathbf{D}}' \mathbf{P} \mathbf{e}_m \mathbf{e}_m^T = \frac{\tilde{\mathbf{D}}'_{ll}}{\tilde{\mathbf{D}}_{ll}} \tilde{\mathbf{A}}_{lm} \mathbf{e}_m^T. \tag{59}$$

Now, for case (1), we have $\tilde{\mathbf{A}}_{lm} = 1$ and $\tilde{\mathbf{D}}'_{ll} = \tilde{\mathbf{D}}_{ll} - 1 \geq 1$. This leads to

$$\mathbf{e}_l^T \tilde{\mathbf{D}}' |\mathbf{P} - \mathbf{P}'| \mathbf{e}_m \mathbf{e}_m^T = \frac{\tilde{\mathbf{D}}'_{ll}}{\tilde{\mathbf{D}}_{ll}} \tilde{\mathbf{A}}_{lm} \mathbf{e}_m^T = \frac{\tilde{\mathbf{D}}_{ll} - 1}{\tilde{\mathbf{D}}_{ll}} \mathbf{e}_m^T. \tag{60}$$

For case (2), we clearly have $\mathbf{e}_l^T \tilde{\mathbf{D}}' |\mathbf{P} - \mathbf{P}'| \mathbf{e}_m \mathbf{e}_m^T = 0$ as $\tilde{\mathbf{A}}_{lm} = 0$.

Hence, for each $j \neq m$ and under the setting in case (1), $\mathbf{e}_l^T \tilde{\mathbf{D}}' |\mathbf{P} - \mathbf{P}'|$ equals the row vector

$$\left[\frac{\tilde{\mathbf{A}}_{l1}}{\tilde{\mathbf{D}}_{ll}}, \frac{\tilde{\mathbf{A}}_{l2}}{\tilde{\mathbf{D}}_{ll}}, \cdots, \frac{\tilde{\mathbf{A}}_{l(m-1)}}{\tilde{\mathbf{D}}_{ll}}, 0, \frac{\tilde{\mathbf{A}}_{l(m+1)}}{\tilde{\mathbf{D}}_{ll}}, \cdots\right] + \left[0, \cdots, 0, \frac{\tilde{\mathbf{D}}_{ll} - 1}{\tilde{\mathbf{D}}_{ll}}, 0, \cdots\right], \tag{61}$$

where the $m^{th}$ entry of the first row vector equals 0 and the second row vector is all-zeros except for the $m^{th}$ entry. Note that the first row vector times $\frac{\tilde{\mathbf{D}}_{ll}}{\tilde{\mathbf{D}}_{ll}-1} > 1$ is a probability vector. Hence, by the property of $\mathbf{P}^{K-k}$ being a right-stochastic matrix, we have

$$\left[\frac{\tilde{\mathbf{A}}_{l1}}{\tilde{\mathbf{D}}_{ll}}, \frac{\tilde{\mathbf{A}}_{l2}}{\tilde{\mathbf{D}}_{ll}}, \cdots, \frac{\tilde{\mathbf{A}}_{l(m-1)}}{\tilde{\mathbf{D}}_{ll}}, 0, \frac{\tilde{\mathbf{A}}_{l(m+1)}}{\tilde{\mathbf{D}}_{ll}}, \cdots\right] \mathbf{P}^{K-k} \mathbf{1} \leq 1. \tag{62}$$

Since $\frac{\tilde{\mathbf{D}}_{ll}-1}{\tilde{\mathbf{D}}_{ll}} < 1$, we also have

$$\left[0, \cdots, 0, \frac{\tilde{\mathbf{D}}_{jj} - 1}{\tilde{\mathbf{D}}_{jj}}, 0, \cdots\right] \mathbf{P}^{K-k} \mathbf{1} \leq 1. \tag{63}$$

Together, this shows that for each $j \neq m$ and for the case (1), one has

$$\mathbf{e}_l^T \tilde{\mathbf{D}}' |\mathbf{P} - \mathbf{P}'| \mathbf{P}^{K-k} \mathbf{1} \leq 2. \tag{64}$$

For case (2), note that $\mathbf{e}_l^T \tilde{\mathbf{D}}' |\mathbf{P} - \mathbf{P}'|$ is an all-zero row vector. Note also that, excluding self-loops, there are at most $\tilde{\mathbf{D}}_{mm} - 1$ neighbors $l$ of $m$ (case (1)). Thus,

$$\sum_{l \neq m} \mathbf{e}_l^T \tilde{\mathbf{D}}' |\mathbf{P} - \mathbf{P}'| \mathbf{P}^{K-k} \mathbf{1} \leq 2\tilde{\mathbf{D}}_{mm} - 2. \tag{65}$$

To conclude the proof, we analyze the term $l = m$. For any $i \in [n]$, by Proposition A.6 we have

$$\begin{aligned}
&\mathbf{e}_m^T \tilde{\mathbf{D}}' |\mathbf{P} - \mathbf{P}'| \mathbf{e}_r \mathbf{e}_r^T \\
&= \mathbf{e}_m^T \tilde{\mathbf{D}}' \mathbf{P} \mathbf{e}_r \mathbf{e}_r^T = \frac{\tilde{\mathbf{D}}'_{mm}}{\tilde{\mathbf{D}}_{mm}} \tilde{\mathbf{A}}_{mr} \mathbf{e}_r^T \overset{(a)}{=} \frac{\tilde{\mathbf{A}}_{mr}}{\tilde{\mathbf{D}}_{mm}} \mathbf{e}_r^T = \mathbf{e}_m^T \tilde{\mathbf{D}}^{-1} \tilde{\mathbf{A}} \mathbf{e}_r \mathbf{e}_r^T = \mathbf{e}_m^T \mathbf{P} \mathbf{e}_r \mathbf{e}_r^T \tag{66}
\end{aligned}$$

where (a) holds by definition, and since $\tilde{\mathbf{D}}'_{mm} = 1$. Thus,

$$\mathbf{e}_m^T \tilde{\mathbf{D}}' |\mathbf{P} - \mathbf{P}'| \mathbf{P}^{K-k} \mathbf{1} = \mathbf{e}_m^T \mathbf{P} \mathbf{P}^{K-k} \mathbf{1} = \mathbf{p}^T \mathbf{1} = 1, \tag{67}$$

for some probability vector $\mathbf{p}$. We have hence shown that for any $k \in [K]$,

$$\sum_{j=1}^n \mathbf{e}_j^T \tilde{\mathbf{D}}' |\mathbf{P} - \mathbf{P}'| \mathbf{P}^{K-k} \mathbf{1} \leq 2\tilde{\mathbf{D}}_{mm} - 2 + 1 = 2\tilde{\mathbf{D}}_{mm} - 1. \tag{68}$$

This completes the proof. $\qquad\square$

## A.16 PROOF OF LEMMA A.7

**Lemma.** *Assume that $\|\mathbf{e}_i^T \mathbf{S}\| \leq 1$, $\forall i \neq m$ and that $\mathbf{e}_m^T \mathbf{S} = \mathbf{0}^T$. Then $\forall i \in [n]$, $K \geq 0$, we have $\|\mathbf{e}_i^T (\mathbf{P}')^K \mathbf{S}\| \leq 1$, where $\mathbf{P} = \tilde{\mathbf{D}}^{-1} \tilde{\mathbf{A}}$ and $\mathbf{P}' = (\tilde{\mathbf{D}}')^{-1} \tilde{\mathbf{A}}'$.*

*Proof.* The proof is similar to the proof of Lemma A.1, and based on induction. The base case $k = 0$ is obviously true by assumption. Now, assume that the claim is true for $k = K - 1$ and let

$\mathbf{Z}^{(K-1)} = (\mathbf{P}')^{(K-1)}\mathbf{S}$. Then, $\forall i \neq m$,

$$\|\mathbf{e}_i^T(\mathbf{P}')^K\mathbf{S}\| = \|\mathbf{e}_i^T\mathbf{P}'\mathbf{Z}^{(K-1)}\| = \|\frac{1}{\tilde{\mathbf{D}}_{ii}'}\sum_{j:\tilde{\mathbf{A}}_{ij}'=1}\mathbf{e}_j^T\mathbf{Z}^{(K-1)}\|$$

$$\leq \frac{1}{\tilde{\mathbf{D}}_{ii}'}\sum_{j:\tilde{\mathbf{A}}_{ij}'=1}\|\mathbf{e}_j^T\mathbf{Z}^{(K-1)}\|$$

$$\overset{(a)}{\leq} \frac{1}{\tilde{\mathbf{D}}_{ii}'}\sum_{j:\tilde{\mathbf{A}}_{ij}'=1}1$$

$$\leq \frac{1}{\tilde{\mathbf{D}}_{ii}'}\sum_{j:\tilde{\mathbf{A}}_{ij}'=1}1 = \frac{1}{\tilde{\mathbf{D}}_{ii}'}\tilde{\mathbf{D}}_{ii}' = 1. \tag{69}$$

Here, (a) is due to our hypothesis for $k = K - 1$. For $i = m$, note that $\tilde{\mathbf{A}}_{mj}' = 0$, $\forall j \in [n]$. Thus, $\|\mathbf{e}_n^T(\mathbf{P}')^K\mathbf{S}\| = 0 \leq 1$. This completes the proof. $\qquad\square$

## A.17  PROOF OF LEMMA A.8

**Lemma.** *Assume that* $\|\mathbf{e}_i^T\mathbf{S}\| \leq 1$, $\forall i \in [n]$. *Then,* $\forall i \in [n]$, $K \geq 0$, *we have* $\|\frac{1}{\sqrt{K+1}}\mathbf{e}_i^T[\mathbf{S}, \mathbf{PS}, \mathbf{P}^2\mathbf{S}, \cdots, \mathbf{P}^K\mathbf{S}]\| \leq 1$, *where* $\mathbf{P} = \tilde{\mathbf{D}}^{-1}\tilde{\mathbf{A}}$.

*Proof.* By Lemma A.1, we have $\|\mathbf{e}_i^T\mathbf{P}^k\mathbf{S}\| \leq 1$, $\forall k \in [K]$. Thus,

$$\|\frac{1}{\sqrt{K+1}}\mathbf{e}_i^T[\mathbf{S}, \mathbf{PS}, \mathbf{P}^2\mathbf{S}, \cdots, \mathbf{P}^K\mathbf{S}]\|^2 = \frac{1}{K+1}\left(\sum_{k=0}^K\|\mathbf{e}_i^T\mathbf{P}^k\mathbf{S}\|^2\right) \leq 1, \tag{70}$$

which complete the proof. $\qquad\square$

*Remark.* Using the normalization $\frac{1}{K+1}$ also leads to a norm bounded by 1. Hence, the norm of each row of $\mathbf{Z}$ is bounded by 1. We need the normalization $\frac{1}{K+1}$ instead of $\frac{1}{\sqrt{K+1}}$ to accommodate another claim in the proof.

## A.18  PROOF OF PROPOSITION A.6

**Proposition.** *We have* $\mathbf{e}_i^T|\mathbf{P} - \mathbf{P}'|\mathbf{e}_j = \mathbf{e}_i^T(\mathbf{P}' - \mathbf{P})\mathbf{e}_j$, $\forall i, j \neq m$. *For* $i = m$ *or* $j = m$, $\mathbf{e}_i^T|\mathbf{P} - \mathbf{P}'|\mathbf{e}_j = \mathbf{e}_i^T\mathbf{P}\mathbf{e}_j$.

*Proof.* For the first case when $\forall i, j \neq m$,

$$\mathbf{e}_i^T(\mathbf{P} - \mathbf{P}')\mathbf{e}_j = \frac{\tilde{\mathbf{A}}_{ij}}{\tilde{\mathbf{D}}_{ii}} - \frac{\tilde{\mathbf{A}}_{ij}'}{\tilde{\mathbf{D}}_{ii}'}. \tag{71}$$

Recall that by definition, in this case we have $\tilde{\mathbf{A}}_{ij} = \tilde{\mathbf{A}}_{ij}'$. Now, there are two cases to consider: (1) $i$ is a neighbor of $m$; (2) $i$ is not a neighbor of $m$. For (1), we know that $\tilde{\mathbf{D}}_{ii}' = \tilde{\mathbf{D}}_{ii} - 1 \geq 1$. As a result,

$$\frac{\tilde{\mathbf{A}}_{ij}}{\tilde{\mathbf{D}}_{ii}} - \frac{\tilde{\mathbf{A}}_{ij}'}{\tilde{\mathbf{D}}_{ii}'} = \frac{\tilde{\mathbf{A}}_{ij}}{\tilde{\mathbf{D}}_{ii}} - \frac{\tilde{\mathbf{A}}_{ij}}{\tilde{\mathbf{D}}_{ii} - 1} < 0. \tag{72}$$

This directly implies $\mathbf{e}_i^T|\mathbf{P} - \mathbf{P}'|\mathbf{e}_j = \mathbf{e}_i^T(\mathbf{P}' - \mathbf{P})\mathbf{e}_j$. For (2), we know that $\tilde{\mathbf{D}}_{ii}' = \tilde{\mathbf{D}}_{ii}$ and thus $\mathbf{e}_i^T|\mathbf{P} - \mathbf{P}'|\mathbf{e}_j = 0 = \mathbf{e}_i^T(\mathbf{P}' - \mathbf{P})\mathbf{e}_j$. These claims complete the proof for the first part. For the case that $i = m$ or $j = m$, note that since both the $m^{th}$ row and column are all-zeros for $\mathbf{P}'$, we simply have $\mathbf{e}_i^T|\mathbf{P} - \mathbf{P}'|\mathbf{e}_j = \mathbf{e}_i^T\mathbf{P}\mathbf{e}_j$. Note that in establishing the claim we also used the fact that $\mathbf{P}$ is nonnegative. This completes the proof. $\qquad\square$

Table 1: Properties of benchmarking datasets.

| Name | #nodes | #edges | #features | #classes | train/val/test |
|------|--------|--------|-----------|----------|----------------|
| Cora | 2,708 | 10,556 | 1,433 | 7 | 1,208/500/1,000 |
| Citeseer | 3,327 | 9,104 | 3,703 | 6 | 1,827/500/1,000 |
| PubMed | 19,717 | 88,648 | 500 | 3 | 18,217/500/1,000 |
| Computers | 13,752 | 491,722 | 767 | 10 | 12,252/500/1,000 |
| Photo | 7,650 | 238,162 | 745 | 8 | 6,150/500/1,000 |
| ogbn-arxiv | 169,343 | 1,166,243 | 128 | 40 | 90,941/29,799/48,603 |

## A.19 ADDITIONAL EXPERIMENTAL DETAILS

All our experiments were executed on a Linux machine with 48 cores, 376GB of system memory, and two NVIDIA Tesla P100 GPUs with 12GB of GPU memory each. Information about all datasets can be found in Table 1. The data split is public and obtained from PyTorch Geometric Fey & Lenssen (2019). We used the "full" split option for Cora, Citeseer and Pubmed. Since there is no public split for Computers and Photo, we adopted a similar setting as for the citation networks via random splits (i.e., 500 nodes in the validation set and 1,000 nodes in the test set). The data split for ogbn-arxiv is the public split provided by the Open Graph Benchmark Hu et al. (2020).

**Dependency on the node degree.** We verified our Theorem 4.5 and Theorem 4.6 for node degree dependencies on Photo, Cora, Citeseer and Pubmed. The results are presented in Figure 5.

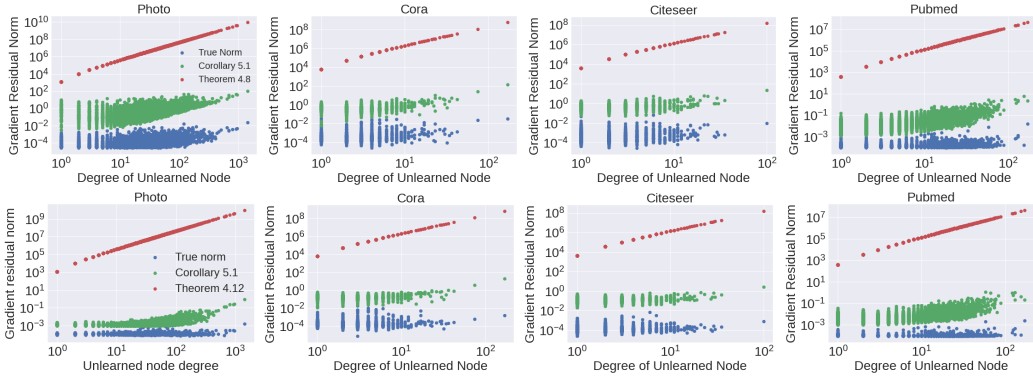

Figure 5: Additional examination of the degree dependency result from Theorem 4.5 (top) and Theorem 4.6 (bottom).

**Nonaccumulative time for each of the unlearning procedures.** Figure 6 shows the average time complexity for each unlearning step on the Cora dataset. The spikes for approximate graph unlearning methods and Unstructured Unlearning (Guo et al., 2020) corresponds to retraining after a removal.

**Membership inference attacks for unlearned models.** We performed experiments for node unlearning tasks and applied the membership inference attack for GNNs reported in Olatunji et al. (2021b) to our obtained updated models. For simplicity, we used the Cora dataset and removed up to 100 nodes. After each removal, we applied an MI attack on the updated model. We compare the results of our *SGC node unlearning* approach with that of the *original SGC model without updates*, which is the model trained on the full dataset, and with *SGC retraining*, which corresponds to the model obtained after retraining upon each removal request. We repeated the experiments with 10 different trails and random splits and averaged the results. As shown in Figure 7, even for full *SGC retraining* the attack model can still identify parts of the removed nodes in the training set, and the result of *SGC node unlearning* is slightly worse (w.r.t privacy) than retraining since our method is concerned with approximate unlearning. Note that the performance of the MI attack on the original model is consistent with the results from Olatunji et al. (2021b) and significantly worse than both our unlearning as well as the complete retraining method. This, from the experimental side, shows that

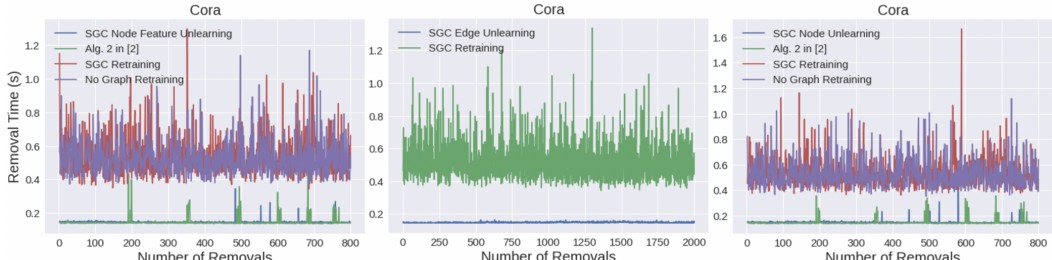

Figure 6: Nonaccumulative time for each removal step on the Cora dataset. The setting is the same as in Figure 2.

our method offers similar privacy-preserving performance as full retraining, and better performance when compared to the original model without unlearning. Nevertheless, the results also motivate the search for alternatives to MI attacks for unlearning schemes.

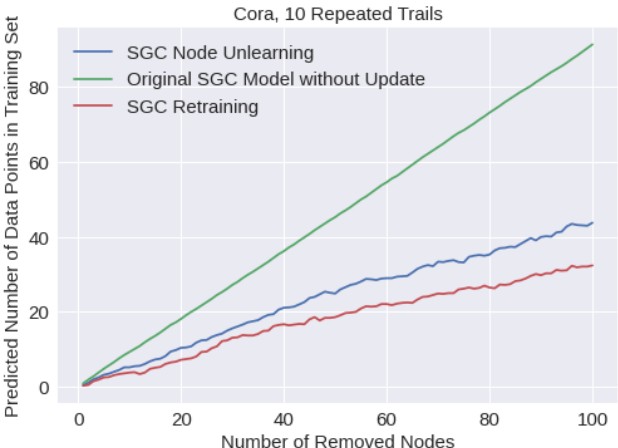

Figure 7: The number of data points predicted by the attack model to lie in the training set. The setting is the same as in Figure 2.

**Additional experiments.** The performance of our proposed approximate graph unlearning methods on three datasets, including Citeseer, Pubmed and Amazon Photo, is shown in Figure 8. It is worth pointing out that our bound on the gradient residual norm in Section 4 does not guarantee the generalization ability of the updated model. Therefore, It could happen that the test accuracy increases as we remove information from the training set, as shown in the second row of Figure 8, or that the performance is not very stable, as seen in the third row of Figure 8.

We also performed additional experiments on the Cora dataset, with results shown in Figure 9. The first row shows the average performance over 10 repeated trails with random splitting, and the conclusion is the same as the one stated in Section 6. The second row shows the performance on GPR-based models. Note that when the number of removal requests becomes large, the performance of GPR-based models degrades much faster than that of SGC-based models. This observation is consistent with our discussion of GPR-based models. None of the retraining methods involves noise.

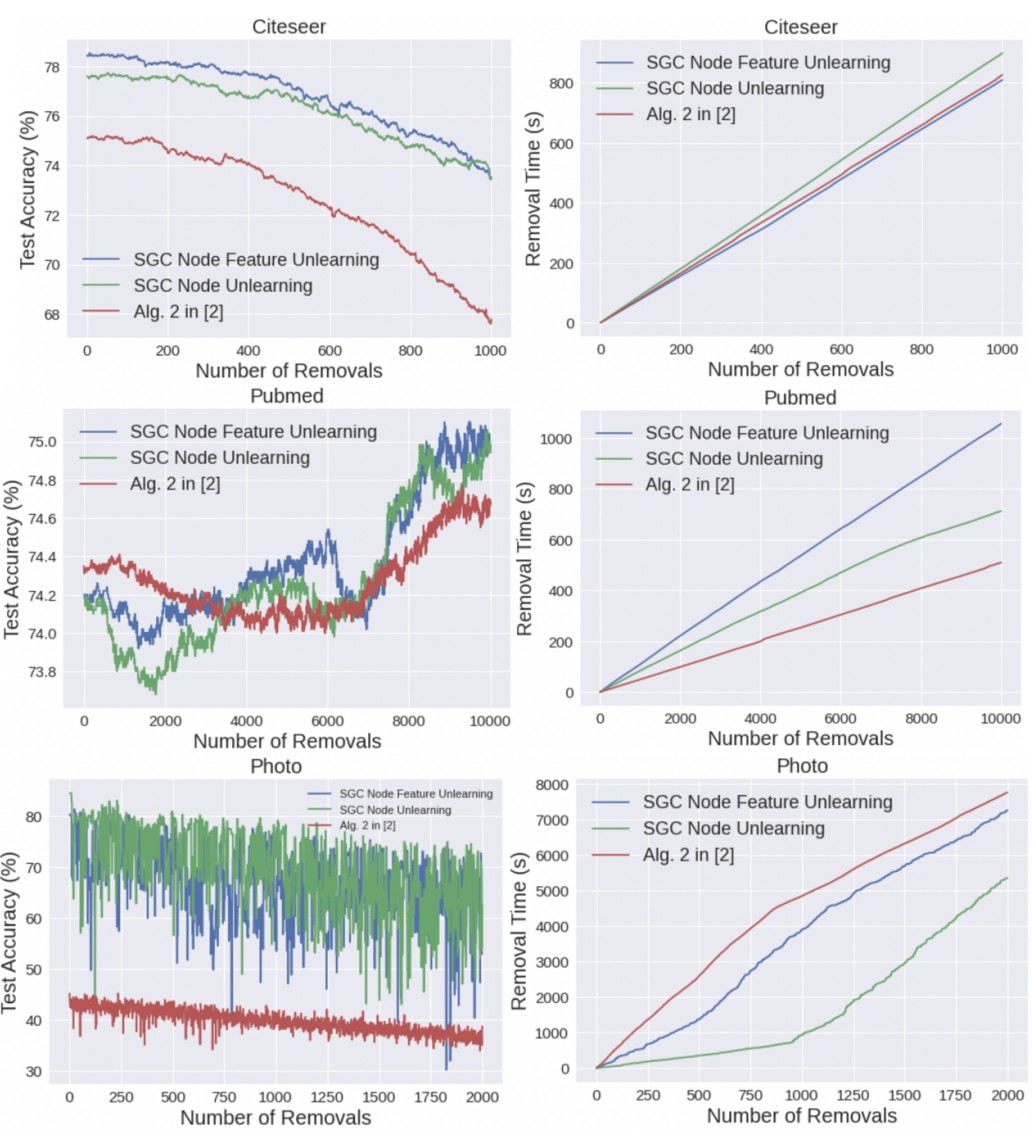

Figure 8: Performance of approximate graph unlearning methods on different datasets. First Row: We removed up to $55\%$ of the training data in Citeseer. Second Row: We removed up to $50\%$ of the training data in Pubmed. Third Row: We set $\alpha = 10, \lambda = 10^{-4}$, and removed up to $30\%$ of the training data in Amazon Photo.

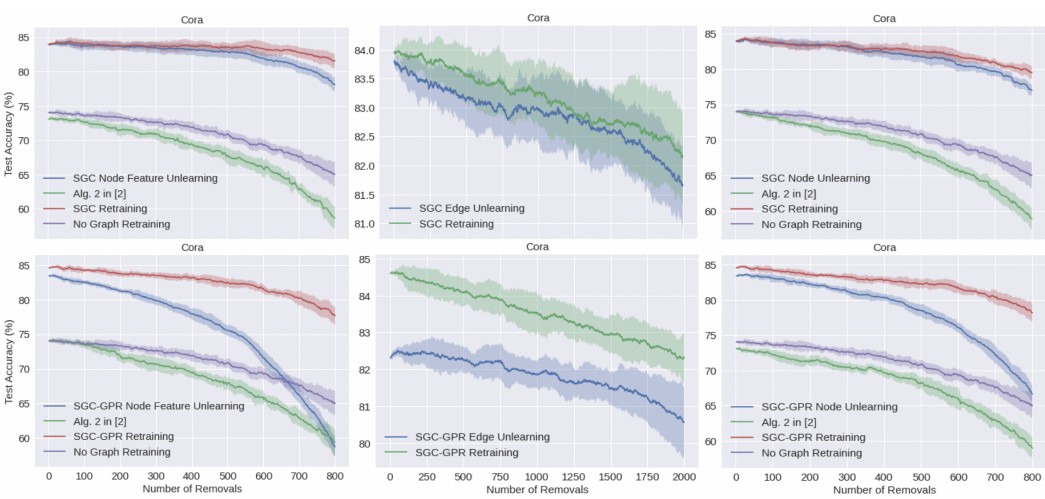

Figure 9: Performance of approximate graph unlearning methods on Cora. First Row: The reported statistics are based on averaging over 10 repeated trails with random splitting. Second Row: GPR-based models are used to obtain node embedding. All other settings are the same as in Figure 2.

