# OpenReview forum: "Efficient Model Updates for Approximate Unlearning of Graph-Structured Data"
_ICLR.cc/2023/Conference — ICLR 2023 poster_

### Official Review · Reviewer_dA7m · 2022-10-24

**Confidence:** 3
**Correctness:** 3
**Technical Novelty And Significance:** 4
**Empirical Novelty And Significance:** 4
**Recommendation:** 6

**Clarity, Quality, Novelty And Reproducibility:**

I found this paper to be well-written despite some flaws that hindered the reading.

- Privacy cost: At the end of the introduction (page 2), the reader (except if she/he is familiar with approximate unlearning) has no idea what privacy cost is and the given value makes no sense to her/him. The same occurs in section 2 about DP-GNNs (page 3). In the definition of certified removal, I guess $e$ denotes $\exp(1)$. Together with the formulation "given $\epsilon>0$ ... where $\epsilon,\delta>0$", it can be confusing: I thought that it was something like "given $e>0$" instead of "given $\epsilon>0$". Beyond this typographical detail, at this point, the authors could elaborate on the formula for what is an acceptable cost and what is too expensive.

- Privacy budget: I may be wrong, but the concept of privacy budget appears on page 7 without any justification, and involves the unknown parameter $\alpha$ (at this point, we only know that it denotes a noise standard deviation without more details). The authors should really develop this part better. It also seems to be related to my question above about the complexity of unlearning vs retraining.

- SGC, GPR, GCN...: these acronyms are not defined in the paper (SGC and GPR are defined in the abstract but should be recalled in the core of the paper, like GNN for example).

- Even if we can guess it, the degree matrix should be defined.

- Typos: "As an demonstrative example" (page 5), "matches that ot Guo et al." (page 5); In Figures 2 and 3, [2] refers to nothing.

In general, the authors should not assume that the reader is very familiar with the paper of Guo et al., 2020.

I also have the following 2 questions/comments:
- Unlike retraining, the unlearning procedure requires to know the data to be removed, which imposes to recompute the model (Simple Graph Convolution or Generalized PageRank) before removing it from the database. I wonder if, in practical applications, this can not be a drawback of the approach.

- What is the idea behind the definition of $\Delta$ (page 5, above eq. (2))? I can see through the given example (node unlearning) that it leads to a similar formula as in Guo et al., 2020. However, in this reference, $\Delta$ is directly defined as this, and does not come from a more general formula. Could the authors comment on this point?


**Strength And Weaknesses:**

This article is an extension of a paper by Guo et al. published in ICML 2020, which investigates approximate unlearning for unstructured data. The present paper extends the methodology and the results to graph data for 2 learning models: Simple Graph Convolution and Generalized PageRank, which is clearly of great interest and certainly non-trivial.

On the substance, I think that this paper suffers from 2 main limitations.
- First, it does not apply to Graph Neural Networks and can therefore appear as quite specific. On this point, it would be interesting for the authors to explain precisely the conceptual limitations of their approach and why it does not apply to more complex models (GCNs for example). However, I understand that this paper is a first step towards approximate unlearning for graphs and can not handle all models at once.
- Second, it does not investigate from a theoretical perspective the time-complexity of their removal procedure, while the main objective is to save computation time compared to retraining. It would be interesting to know the theoretical gain in each of the cases considered, also in the case of successive requests.

On the form, my comments are given in the following box.


**Summary Of The Paper:**

This paper deals with unlearning methods for graphs. It introduces a new algorithm for approximate graph unlearning from different requests (feature, node, and edge unlearning) for Simple Graph Convolution and Generalized PageRank models, and provide theoretical guarantees of unlearning.

**Summary Of The Review:**

I enjoyed reading this paper despite a number of presentation flaws. The approach and results are clearly worthwhile, but the presentation of the paper really needs to be improved to allow for publication.

---

> ### Author Response · Authors · 2022-11-10
> **Response to Reviewer dA7m (1/n)**
>
> We thank Reviewer dA7m for their constructive and helpful comments. Our detailed answers are provided below.
>
> ### Our work cannot directly apply to general Graph Neural Networks (GNNs).
>
> We would like to thank the reviewer for correctly recognizing that our work is the first step towards approximate graph unlearning and that our theoretical contributions are solid. As we mentioned in Appendix A.2, one of our current limitations is that the reported analysis cannot be directly applied to general GNNs. Still, please note that even for unstructured unlearning settings, it is still unclear how to establish a certified unlearning mechanism as those proposed in Guo et al. (2020) for simple neural networks such as MLPs. The only reported approaches for more general neural network unlearning, such as those in Golatkar et al. (2020), are heuristic and do not come with theoretical privacy guarantees. Still, some of our recent findings suggest that it may be possible to adapt the ideas described in our submission using analytical tools from differential privacy (DP), since DP-SGDs ensure the privacy of nonlinear neural networks. The task is highly nontrivial and may take a while to be successfully completed. This is why we chose not to report any preliminary findings regarding general neural networks. Another practically relevant direction is to generalize the heuristic-based approaches in Golatkar et al. (2020) for more powerful GNN architectures, which we are also currently investigating.
>
> ### Theoretical analysis of the time complexity of our unlearning procedure.
>
> The main idea that explains why our unlearning approach and the one proposed by Guo et al. (2020) are more efficient than complete retraining is as follows. Note that the update rule we use is similar to a one-step Newton update, which has a time complexity of $O(nF^2+F^3)$ (Guo et al., 2020), where $n$ stands for the number of nodes and $F$ stands for the feature dimension. The computational bottleneck is the evaluation of the Hessian, which takes $O(nF^2)$ time, and the evaluation of its inverse, which takes $O(F^3)$ time. Note that in most settings, we have $F<n$, so that the time complexity of a single unlearning update equals $O(nF^2)$. Since we are comparing our results to complete retraining, we ignore the complexity of the process used to determine $\mathbf{Z}$, as it is the same for our unlearning approach and complete retraining. The time complexity of retraining depends on the optimizer used. For example, if one uses quasi-Newton’s method (i.e., BFGS) then the time complexity for retraining equals $O(nF^2T)$, where $T$ is the number of steps needed for convergence. Clearly, we can see that our unlearning approach provides savings whenever $T$ is large for each unlearning request. The sequential unlearning approach would just repeat the above argument multiple times provided that we do not run out our privacy budget. We also conducted extensive experiments to establish that the time saving of our unlearning approach compared to retraining is significant (i.e., Figure 2 and 3).

---

> > ### Author Response · Authors · 2022-11-10
> > **Response to Reviewer dA7m (2/n)**
> >
> > ### Explanation of the ``privacy cost’’
> >
> > The privacy cost $(\epsilon,\delta)$ is related to the definition of $(\epsilon,\delta)$-Differential Privacy (DP). It is also related to the definition of “certified” removal in Equation (1). In order to keep the exposition straightforward, and avoid introducing too many mathematical details in the introduction we instead cite the work by Dwork, 2011 for the definition of $(\epsilon,\delta)$-DP in Section 3. We state the formal definition of $(\epsilon,\delta)$-DP as follows:
> >
> >
> > We say a learning algorithm $A$ is $(\epsilon,\delta)$-DP for $\epsilon>0,\delta\in [0,1]$ if for all dataset $\mathcal{D}$ and all dataset $\mathcal{D}^\prime$ that differ on a single element to $\mathcal{D}$, we have
> >
> > $\forall \mathcal{T}\in \mathcal{H},\;\mathbb{P}(A(\mathcal{D})\in \mathcal{T}) \leq \exp(\epsilon)\mathbb{P}(A(\mathcal{D}^\prime)\in \mathcal{T})+\delta,$
> >
> > $\forall \mathcal{T}\in \mathcal{H},\;\mathbb{P}(A(\mathcal{D}^\prime)\in \mathcal{T}) \leq \exp(\epsilon)\mathbb{P}(A(\mathcal{D})\in \mathcal{T})+\delta,$
> >
> > where $\mathcal{H}$ represents a chosen space of models and $\mathcal{T}$ is an arbitrary subset of $\mathcal{H}$.
> >
> >
> > Space permitting, we will try to provide a more rigorous explanation in the main text, and if that is not possible, add significantly more explanation into the current pertinent text in the Supplement. For the suggestion regarding the definition of certified removal, we will change $e^\epsilon$ to $\exp(\epsilon)$ to prevent any ambiguity. Regarding the acceptable cost, we refer to the public document of Apple [1]. The company typically uses $\epsilon<5$ for practical applications involving local differential privacy. Ideally, we would like $\epsilon$ to be as small as possible while ensuring “reasonable” performance.
> >
> > ### Explanation of the ``privacy budget’’
> >
> > We apologize for the confusion. The privacy budget is based on the setup of Theorem 4.1 and it also appears in Algorithm 2 (line 16) of the work by Guo et al. (2020). Note that the authors of the latter work use $\sigma$ for the standard deviation while we use $\alpha$. The constant $c$ therein is replaced by $c_0$ in our work. This budget is used to characterize the quantity of gradient residual norm we can tolerate in order to achieve $(\epsilon,\delta)$-certified removal. Please check our Algorithm 2 in Appendix A.6 for details regarding how to use the privacy budget in sequential unlearning. Roughly speaking, for each unlearning request, we compute the data-dependent bound as stated in Corollary 5.1 and record the accumulated value for the whole sequence of requests. Then we check whether it exceeds our predefined privacy budget. If so, we retrain the model. If not, we proceed with the next unlearning request.
> >
> > ### Some acronyms are not defined or should be recalled.
> >
> > We thank the reviewer for their suggestion. We will address this issue in our revision.
> >
> > ### Degree matrix should be defined.
> >
> > We have defined the degree matrix $\mathbf{D}$ in Section 3, line 6 of the paragraph pertaining to the notation used in the manuscript. We also defined the self-loop added version of it $\tilde{\mathbf{D}}$ in Section 4.1, the fifth line of the first paragraph. We will try to emphasize them more in our revision.
> >
> > ### Typos:
> >
> > We thank the reviewer for spotting these typos. The reference [2] is to Guo et al. (2020). We will make all corrections in our revision.
> >
> > ### ``Unlike retraining, the unlearning procedure requires to know the data to be removed…’’
> >
> > We are not sure if we fully understand this question. We tried our best to answer the question based on our understanding/interpretation of the question. Please do let us know if we did not get your question right and we will follow up with further answers.
> >
> > Note that both retraining and our unlearning procedure require recomputing the propagated node features $\mathbf{Z}^\prime$ for the unlearning request, for which we need to have access to the data after removal, $\mathcal{D}^\prime$. The main saving in time here is that unlearning approach can return (approximately) private model parameters with merely one step of Newton’s update while retraining may require several steps for convergence. Also, in case that the reviewer is referring to settings where the data is stored in a distributed manner, both our unlearning and complete retraining would require access to local data after removal as it is necessary for computing the pertinent gradients.

---

> > > ### Author Response · Authors · 2022-11-10
> > > **Response to Reviewer dA7m (3/n,n=3)**
> > >
> > > ### Idea behind the definition of $\Delta$
> > >
> > > Indeed, in the case of unstructured unlearning, the definition of $\Delta$ depends on the gradient of a single data point only. However, in the case of graph unlearning, nodes are correlated through graph propagation and thus the earlier definition of $\Delta$ would not be appropriate. Our motivation for the update rule as well as the analysis of $\Delta$ can be found in Appendix A.3. We state it below for your convenience.
> > >
> > >
> > > Our unlearning mechanism proposed in Section 4 is
> > > $$
> > > \mathbf{w}^{-}=\mathbf{w}^\star + \left[\nabla^2 L(\mathbf{w}^\star,\mathcal{D}^\prime)\right]^{-1}\left[\nabla L(\mathbf{w}^\star,\mathcal{D})-\nabla L(\mathbf{w}^\star,\mathcal{D}^\prime)\right],
> > > $$
> > > and the intuition is stated as follows. Our goal for the updated model is $\nabla L(\mathbf{w}^{-},\mathcal{D}^\prime)=0$. By Taylor series we have that
> > > $$
> > > \nabla L(\mathbf{w}^{-},\mathcal{D}^\prime)\approx \nabla L(\mathbf{w}^\star,\mathcal{D}^\prime) + \nabla^2 L(\mathbf{w}^\star,\mathcal{D}^\prime)(\mathbf{w}^{-}-\mathbf{w}^\star)=0.
> > > $$
> > > Therefore, we have
> > > $$
> > > \mathbf{w}^{-}-\mathbf{w}^\star =\left[\nabla^2 L(\mathbf{w}^\star,\mathcal{D}^\prime)\right]^{-1}\left[0-\nabla L(\mathbf{w}^\star,\mathcal{D}^\prime)\right]
> > > $$
> > > $$
> > > \Rightarrow \mathbf{w}^{-} =\mathbf{w}^\star + \left[\nabla^2 L(\mathbf{w}^\star,\mathcal{D}^\prime)\right]^{-1}\left[\nabla L(\mathbf{w}^\star,\mathcal{D})-\nabla L(\mathbf{w}^\star,\mathcal{D}^\prime)\right].
> > > $$
> > >
> > > The last equality holds due to the fact that $\mathbf{w}^\star$ should be the unique optimizer for the strongly convex loss $L(\mathbf{w},\mathcal{D})$ over the entire dataset $\mathcal{D}$.
> > >
> > >
> > > ## References
> > >
> > > [1] Public differential privacy document of Apple: https://www.apple.com/privacy/docs/Differential_Privacy_Overview.pdf?fbclid=IwAR1cz0ox2zcSwsDrDxqJPua2aZ6AThbTT3TyLWPtINcWDJeX5R7ekanDOgQ

---

> > > > ### Author Response · Authors · 2022-12-08
> > > > **Follow up**
> > > >
> > > > Thank you again for your positive comments. Please feel free to let us know in case you have any further questions. We will try our best to address them.

---

### Official Review · Reviewer_XzFP · 2022-10-25

**Confidence:** 3
**Correctness:** 4
**Technical Novelty And Significance:** 3
**Empirical Novelty And Significance:** 3
**Recommendation:** 6

**Clarity, Quality, Novelty And Reproducibility:**

Clarity
- The figures are difficult to read when printed out. Please use vector images with proper font size.
- The notation e_i is adding extra hurdle to digest the expressions. Subscript (to index matrix entry or data point) would be sufficient in most cases.


**Strength And Weaknesses:**

Strength
- The paper provides a detailed theoretical analysis of the graph unlearning algorithm.
- Graph unlearning is broken down into three subtasks: node feature/edge/node unlearning.

Weaknesses
- The analysis is performed on the simplified convolutional graphs and cannot be generalized to the standard GNN architectures. Although SCG shows competitive performance on some datasets, it is not the most preferable model selection in general. So, it would be good to have some additional experiments with more GNN architectures to show how the proposed approach can be generalized to the other models.
- Following the previous weakness, the assumptions such as the unique optimum of the loss is quite strong are quite strong and cannot meet in general.
- Although it is important to keep similar performance with retraining methods, it is also important to analyze what can be unlearned or not from the unlearning perspective. The current analysis only shows the performance of the membership inference attack and lacks a detailed analysis.
- The proposed method only works when the full dataset is accessible. In many unlearning scenarios, it is not guaranteed to access the full dataset.

**Summary Of The Paper:**

This paper proposes a graph unlearning algorithm with provable theoretical guarantees. The graph unlearning is divided into three sub-unlearning tasks: node feature unlearning, edge unlearning, and node unlearning. A tailored analysis is provided for each subtask. The experiments with six benchmarks show the effectiveness of the proposed method as well as the inherent trade-off between privacy and accuracy.

**Summary Of The Review:**

Although there are some limitations to the proposed approach, it is worth emphasizing that this work lays the foundation of the theoretical analysis of graph unlearning. Some parts of the presentation could be improved further to improved the readability of the manuscript.

---

> ### Author Response · Authors · 2022-11-10
> **Response to Reviewer XzFP**
>
> We thank Reviewer XzFP for their helpful feedback. We address all comments as follows.
>
> ### Our work cannot be directly applied to general Graph Neural Networks (GNNs) and the unique optimum assumption for the linear model is too strong.
>
> We would first like to thank the reviewer for correctly recognizing that our work lays the foundation for the theoretical analysis of graph unlearning problems. We agree that SGCs are not the most desirable model to be used for graph unlearning in general. Ultimately, we would like to establish the unlearning procedure for general GNNs, and in the process remove the unique optimum assumption. To the best of our knowledge, there is no approximate unlearning approach that applies to general GNNs with theoretical guarantees and it is highly nontrivial to get such results. As we mentioned in a related context raised by reviewer dA7m, please note that even for unstructured unlearning settings, it is still unclear how to establish a certified unlearning mechanism as those proposed in Guo et al. (2020) for simple neural networks such as MLPs. The only reported approaches for more general neural network unlearning, such as those in Golatkar et al. (2020), are heuristic and do not come with theoretical privacy guarantees. Still, some of our recent findings suggest that it may be possible to adapt the ideas described in our submission using analytical tools from differential privacy (DP), since DP-SGDs ensure the privacy of nonlinear neural networks. The task is highly nontrivial and may take a while to be successfully completed. This is why we chose not to report any preliminary findings regarding general neural networks. Another practically relevant direction is to generalize the heuristic-based approaches in Golatkar et al. (2020) for more powerful GNN architectures, which we are also currently investigating.
>
> ### Questions on what can be unlearned or not from the unlearning perspective.
>
> We agree this is a very important question and thank the reviewer for pointing this out. Note that we have mentioned in the paragraph right below Corollary 5.1 that we are not able to unlearn nodes with large degrees (i.e., hub nodes). This follows from our Theorem 4.3 and 4.5, which assert that unlearning nodes with large node degrees results in a large gradient residual norm. Hence, we would likely need to retrain from scratch or add more noise during training whenever we receive such unlearning requests. We will further emphasize this important observation in our revision.
>
> ### “The proposed method only works when the full dataset is accessible. In many unlearning scenarios, it is not guaranteed to access the full dataset.”
>
> It is true that in some cases we can only *record* some data statistics after we train the model due to either memory or other resource constraints. This setting is similar to Sekhari et al. (2021) and it would be interesting to see if we can generalize their analysis for graph learning/unlearning. Note that modeling assumptions such as partial graph data availability may be more complicated to handle, since we usually need the entire training graph for inference in both transductive and inductive settings. Still, defining “sufficient statistics” for graph unlearning similar to what was done for the unstructured case (Sekhari et al. 2021) is indeed an interesting future direction.
>
> ### Improve the figure quality with proper font size
>
> Thank you for the suggestion. We will make the required changes in our revision.
>
> ### Using subscript indices for matrices instead of the $\mathbf{e}_i$ notation.
>
> Thank you for the suggestion. We agree that using subscript indices for matrices makes for easier reading of the main text. We find that the $\mathbf{e}_i$ notation makes the proof easier to follow compared to that of subscripts. If the reviewer believes changing the notation will make a big difference, we are going to honor the request.

---

> > ### Author Response · Authors · 2022-12-08
> > **Follow up**
> >
> > Thank you again for your positive comments. Please feel free to let us know in case you have any further questions. We will try our best to address them.

---

### Official Review · Reviewer_VMQZ · 2022-10-26

**Confidence:** 3
**Correctness:** 4
**Technical Novelty And Significance:** 3
**Empirical Novelty And Significance:** 3
**Recommendation:** 8

**Clarity, Quality, Novelty And Reproducibility:**

The work is presented clearly. The overall quality is relatively high. The idea of conducting unlearning algorithms in graph domain is novel. The reproducibility can’t be evaluated due to resource limitation.

**Strength And Weaknesses:**

Strength:

1.	The introduced proposed algorithm and analysis are novel and promising. The process of generalizing unstructured data unlearning analysis to the graph domain is nontrivial. Node embeddings are propagated on graphs during training, and therefore the removal of data is more complex. The theoretical analysis is deep and sound.

2.	The proposed bounds of error are well supported by the experiments. The experimental results support the claim that the unlearning algorithm is faster than retraining. The improvement of accuracy against unstructured algorithm is significant, and the correlation between the accuracy and the degree of removed node makes the bounds more reasonable.

3.	The paper is well-written. Limitation of the work, reasonableness of assumptions and main techniques used in the proof are all well illustrated.

Weakness:

1.	Many of the main techniques in analysis and the algorithm are generalized from Guo’s work, which makes the work less revolutionary.

2.	Most of the analysis is constrained to the binary classification task and linear GNN models, which is not practical. Current GNNs, such as the original GRPGNN, include non-linearity modules. As a result, the applicability of this method in real-world unlearning scenarios could be limited.


**Summary Of The Paper:**

This paper investigates the problem of machine unlearning for graph-structured data. The authors provide an efficient algorithm to approximate graph unlearning in node classification tasks. The algorithm is proved to hold good theoretical properties for specific GNNs (SGC and GPRGNN). The authors also conduct an empirical study to demonstrate that the proposed algorithm outperforms the baselines.

**Summary Of The Review:**

A solid and interesting work that generalizes unlearning algorithms from unstructured data to graph domain with theoretical guarantees. The analysis is comprehensive and well supported by experimental results.

---

> ### Author Response · Authors · 2022-11-10
> **Response to Reviewer VMQZ**
>
> We thank Reviewer VMQZ for their helpful comments and positive feedback. We address all comments as follows.
>
> ### Our work does not directly apply to general Graph Neural Networks (GNNs).
>
> We would like to first thank the reviewer for correctly recognizing our theoretical contribution to the emerging graph unlearning problem. Indeed, we would like to establish unlearning procedure for general GNNs with nonlinearities. But this is a challenging problem to handle. Nevertheless, as mentioned in the manuscript, Appendix A.2, several easier future directions for extending certified graph unlearning for nonlinear models are plausible. As we mentioned in a related context raised by reviewer dA7m, please note that even for unstructured unlearning settings, it is still unclear how to establish a certified unlearning mechanism as those proposed in Guo et al. (2020) for simple neural networks such as MLPs. The only reported approaches for more general neural network unlearning, such as those in Golatkar et al. (2020), are heuristic and do not come with theoretical privacy guarantees. Still, some of our recent findings suggest that it may be possible to adapt the ideas described in our submission using analytical tools from differential privacy (DP), since DP-SGDs ensure the privacy of nonlinear neural networks. The task is highly nontrivial and may take a while to be successfully completed. This is why we chose not to report any preliminary findings regarding general neural networks. Another practically relevant direction is to generalize the heuristic-based approaches in Golatkar et al. (2020) for more powerful GNN architectures, which we are also currently investigating.
>
>
> ### “Many of the main techniques in analysis and the algorithm are generalized from Guo’s work, which makes the work less revolutionary.”
>
> We agree that our approach and one part of the analysis are generalizations of Guo’s work. However, as pointed out by the reviewer themselves, our analysis involves graphs and is hence faced with novel and non-trivial challenges, which require specialized and innovative mathematical derivations.
>
> We would also like to briefly mention the potential impact of our novel analysis pertaining to graph unlearning. We conjecture that our results can be useful in developing more advanced DP-GNNs and unlearning approaches for general GNNs. Our results characterize how changing one node feature/edge/node affects the gradient residual norm, which is related to the notion of sensitivity from the differential privacy literature. We want to emphasize that, to the best of our knowledge, we are the first to provide such an analysis in the context of graph unlearning.

---

> > ### Author Response · Authors · 2022-12-08
> > **Follow up**
> >
> > Thank you again for your positive comments. Please feel free to let us know in case you have any further questions. We will try our best to address them.

---

### Decision · Program_Chairs · 2023-01-20

**Decision:**

Accept: poster

**Justification For Why Not Higher Score:**

Two reviewers' accepts were marginal.

**Justification For Why Not Lower Score:**

All reviewers accepted.

**Metareview: Summary, Strengths And Weaknesses:**

The authors provide an algorithm and analysis for unlearning in certain graph neural networks. The reviewers found that the paper addresses only a subset of graph neural networks, in particular does not address nonlinearities; but, builds upon prior work in unlearning for MLPs to provide a solid initial foray into the theoretical basis of graph learning, and provides solid empirical evidence to support the performance of the proposed algorithm.

**Note From Pc:**

if the above contains the word "oral" or "spotlight" please see: "oral" presentation means -> notable-top-5% and "spotlight" means -> notable-top-25%. As stated in our emails, we are disassociating presentation type from AC recommendations